# Correcting market failure for no-regret electric road investments under uncertainty

Jakob Rogstadius ®[1] ✉, Hampus Alfredsson ®[1], Henrik Sällberg[2] & Karl-Filip Faxén[1]

Several electric road system technologies that enable in-motion charging of electric vehicles are nearing market readiness. However, substantial contribution to decarbonization requires rapid deployment on an international scale. Investment is discouraged by prior research that has identified that declining battery costs may eventually leave the infrastructure a stranded asset. We explore under what circumstances electric roads offer effective and low-risk decarbonization of European heavy-duty road freight. Transport system dynamics are explored and quantified, through pairwise comparison of scenarios with and without electric road incorporation, using a purpose-built agent-based simulation (MOSTACHI). Prior stranded asset risks are confirmed, but we show that policy that encourages high electric road utilization can correct for market failures and make the infrastructure a no-regret investment in much of Europe – never yielding worse outcomes than not investing. Electric roads are shown to be an effective risk mitigation strategy, achieving market-driven phase-out of fossil fuels before 2050, also in pessimistic scenarios where static charging alone would be insufficient. Electric roads reduce levelized system cost by 0–17%, greenhouse gas emissions by 7–63% (2030 to 2050, cumulative) and battery mineral demand by 20–40%. Benefits are maximized with early and predictable deployment.

The European Union (EU) has committed to a legally binding target of reducing net greenhouse gas (GHG) emissions by 55% by 2030, compared to 1990 levels, and to net zero by 2050. Furthermore, the European Scientific Advisory Board on Climate Change advises a 90–95% by 2040[1]. While emissions from most other sectors have decreased, road transport emissions have remained around 30% above 1990 levels since 2007. Transport alone threatens to exceed the EU's entire carbon budget[2]. Heavy-duty vehicles (HDVs) are notably problematic, with current policy and technologies leading only towards a 64% reduction by 2050[3].

This study contributes to a growing body of research into if electric road systems (ERS) should be deployed along major European routes to accelerate the decarbonization of HDVs. ERS technologies allow for charging of battery-electric vehicles while in motion, either through conductive or wireless power transfer. Such technologies are

being tested worldwide[4], with large-scale (>500 km) installations being considered by public or private entities in multiple countries, including China, India, the United States, the Netherlands, and France.

Selection of freight solutions is primarily driven by cost minimization[5,6]. Prior studies indicate that ERS can reduce systemic costs[7] by enabling travel with smaller batteries, thereby decreasing vehicle costs and weight[8–12]. However, for ERS infrastructure to be cost-effective, significant economies of scale (high traffic participation) and scope (large geographic coverage) are essential[9,10]. Sensitivity analysis looking at hundreds of future scenarios has indicated high ERS participation is probable, but also identified a risk that cheap batteries will make overnight charging more competitive, causing a vicious cycle of reduced ERS usage and rising costs per remaining user[13]. If fears of stranded assets cannot be settled, ERS is unlikely to play a significant role in transport decarbonization.

[1]RISE Research Institutes of Sweden AB, Göteborg, Sweden. [2]Blekinge Institute of Technology, Karlskrona, Sweden. ✉e-mail: jakob.rogstadius@ri.se

We hypothesize that this negative feedback loop can be broken by policy instruments that incentivize high ERS utilization, and inspired by a study of the relationship between ERS pricing and utilization[14], we evaluate if pairing ERS with such policy (e.g., a price cap on ERS charging) would make the infrastructure socio-economically and environmentally advantageous at system level even under ERS-adverse future conditions.

We test this hypothesis experimentally using Model for Optimization and Simulation of Traffic And CHarging Infrastructure (MOSTACHI), a simulation tool purpose-built to study interaction effects, feedback loops, and emerging behavior among logistics and charging infrastructure in space and over time. In MOSTACHI, transport electrification and charging infrastructure expansion are jointly driven by local decisions to electrify transport along individual routes when this reduces levelized transport costs below diesel propulsion. See "Methods" for details. MOSTACHI can be applied to other regions, e.g., by using available road freight patterns between all European NUTS3 (Nomenclature of Territorial Units for Statistics) regions[15].

Prior work has generally used fixed input assumptions, without capturing several complex dynamics present in the system. Our analysis covers trucks weighing 16–60 tons under average European conditions, simulated across 200,000 overlapping transport routes on the Swedish road network. Notably, as Sweden has some of Europe's sparsest traffic, simulated traffic has been scaled to 2.5× Swedish densities. This brings our simulations close to European averages. Traffic on top European roads is up to 3.5× denser than on top Swedish roads.

Our experiments quantify the (1) ranges of outcomes in system cost, GHG emissions, and other key indicators, (2) substitution effects between ERS and three types of static charging, and (3) ERS impact on battery demand per vehicle and at the fleet level. We conclude that investment in an ERS network is a no-regret choice, if supported by policy, and that earlier investment improves the expected return. We therefore call for rapid integration of ERS in key European policy instruments, e.g., AFIR (Alternative Fuels Infrastructure Regulation, due for revision in 2026) and coordinated national planning for ERS on the main European road network. Limitations of the study are discussed in "Study design limitations and implications" and "Model limitations".

## Results

### Experimental design

Five main conditions were defined to assess the short- and long-term viability of ERS under expected and extreme future conditions. The five conditions (Table 1) were defined based on dominant system dynamics identified through Monte Carlo analysis, discussed later. Other realistic future scenarios are therefore expected to yield outcomes between our neutral and extreme scenarios. The conditions include: (1) Neutral, with standard parameters, (2) Trucks Driven More, with increasing daily vehicle use, (3) ERS Unfavorable, with higher daytime electricity and ERS infrastructure costs and round-the-clock use of public static charging, (4) Static-then-ERS, with accelerated construction of public static charging and delayed ERS construction, and (5) Capped Static, with reduced availability of static charging.

Trucks Driven More represents possibilities for increased multi-shift or autonomous vehicle operation, possibly facilitated by ERS. While driver costs have not been altered, reducing them would amplify this scenario. Capped Static is included due to a lack of evidence that ubiquitous static charging is realistic—the last remaining sites could have physical conditions or utilization patterns that make installation uneconomical. ERS Unfavorable represents the inherent risks and opportunities when deploying new types of infrastructure, combined with recent years' price turbulence in the European electricity market. Static-then-ERS evaluates what can be achieved with raised short-term ambition but delayed decisions on ERS. Which scenario best represents the future will likely not become apparent before 2035, while many decisions on ERS are likely to be taken under uncertainty before 2030.

Within each of the five conditions, five scenarios are compared, for a total of 25 scenarios: without ERS, with ERS networks spanning up 2000 or 6000 km (bidirectional, including gaps in coverage), and with or without policies encouraging ERS use. These road networks approximately correspond in scope to the Trans-European Transport Network (TEN-T) Core and Comprehensive networks, within the populated parts of Sweden. Two thousand kilometers is approximately what has been discussed at political levels in Sweden. The 25 main scenarios are used to understand what impact the addition of ERS to the charging mix would have on overall system cost, GHG emissions, transport electrification rates, energy carrier demand, and total demand for charging of each type.

ERS-incentivizing policy was simulated by capping the infrastructure-related component of the ERS charging fee to the average levelized cost of public static fast charging stations. This represents a subsidy if the user base is insufficient to cover the full ERS cost and otherwise has no effect. Diesel propulsion is always retained as an option.

Table 1 lists model parameter variations in each condition. ERS coverage is always 40%, i.e., for 100 km of bidirectional road within the ERS network, 40 km has ERS infrastructure, to yield a distance-averaged maximum power of 150 kW per vehicle (40% of 375 kW). Above this

**Table 1 | Scenario definitions**

| Condition | Global model parameters | Charging infrastructure |
|---|---|---|
| Neutral | All parameters have their default values (Supplementary Tables 1–13). | At depots: build period 2025–2050, capped at 95% of possible locations; maximum 150 kW per truck. At destinations: build period 2030–2045; max. 30% of locations, max. 500 kW. At rest stops: build period 2025–2045, max 100% of locations, max. 1500 kW. ERS: build period 2030 onwards; max. 1500 km (skipping gaps) built per 5-year period; [0, 2000, 6000] km road network with max. 40% coverage; max. 375 kW. |
| Truck Driven More | Vehicle daily time in transit and in use increases annually by 2% (from 9 to 11 and 12–15 h in 2020). Daily time at depot and annual mileage are adjusted accordingly. Total transport work is unaffected, i.e., the vehicle stock implicitly shrinks. Infrastructure utilization rates decrease by 1% annually for depot chargers and increase by 1% for the remaining types. | |
| ERS Unfavorable | 2× daytime electricity cost. 2× ERS construction cost. 0.5x annual traffic volume increase rate (1% from 2%). Utilization rates of depot and rest stop charging infrastructure have increased from 43% to 63% to simulate the effects of co-locating the two. | |
| Static-then- ERS | As "Neutral." | Depot charging is unchanged. Destination charging 2030–2035. Rest stop charging 2025–2035. ERS from 2040. |
| Capped Static | As "Neutral." | Static charging is capped at 65% of depots, 15% of destinations, and 65% of rest stops. |

The main experiments in the study, presented in Figs. 1, 3, 4 and 6 below, compare outcomes under four sets of global model parameters, with each condition further split into five scenarios, with three Electric Road System (ERS) conditions and two policy conditions.

distance-averaged power, prior work observed no significant effects on ERS utility[16].

Charging infrastructure installation rates for the first three conditions are set to be highly ambitious but not impossible in the EU. In Static-then-ERS's, expansion rates for static charging are optimistic even for Northern Europe. Long-term feasibility of static charging expansion is estimated at 95% of overnight stops ("depots"), 100% of daytime rest stops, and 30% of unloading sites ("destinations"). These penetration rates assume adaptations of both power grids and logistics. As these assumed upper limits to availability of static charging are not evidence-based, we also include Capped Static, with lower maximum penetration rates.

Maximum charging powers are 150 kW for depots (enough to fully charge the largest batteries in 10 h), 1500 kW at rest stops (enough to charge 75% of the largest batteries in 45 min), and 500 kW at destinations (maximum expected capacity). ERS is capped at 375 kW; approximately the upper technical limit today for conductive ground-based technologies. Simulated charging powers are maximum per-vehicle energy transfer rates averaged over the duration of a charging session and may be capped by vehicle capabilities—e.g., 1500 kW charging requires large battery packs or future battery technology.

Simulated battery-electric truck (BET) traction battery capacity options are $75_{16}$, $150_{16,24}$, 250, 450, 700, $1000_{24,40,60}$, and $1500_{40,60}$ kWh, with subscripts indicating options for subsets of truck tonnages. The smallest capacities are prohibited in early years if the resulting maximum output power is below current diesel truck specifications. Capacities above those simulated have experimentally been determined to be unattractive. Global supply of renewable drop-in fuels (biofuels or e-fuels) is limited in reality and therefore capped in the simulation at 5% of total 2020 fuel consumption (approximately current EU consumption), with supply assumed to double by 2050 (2% annual increase). The renewable share of total diesel consumption is dynamically adjusted to maximize the use of this limited supply.

The simulation spans from 2020 to 2054, reflecting infrastructure investment horizons, in 5-year intervals. Periods are referred to using the first year. Electrification begins in 2025.

A separate dataset in a factorial design ($n = 513$ scenarios, $n = 54$ step changes per charging infrastructure type) is used to explain marginal changes in outcomes from marginal changes in charging infrastructure availability. The factorial design uses global model parameter assumptions from the Neutral condition and contains all combinations of three availability levels for each type of static charging (0%, 25% and 75% of possible locations), three ERS network sizes (0 km, 2000 km, 6000 km), three ERS coverage ratios (25%, 50%, 100%) and three ERS power levels (up to 100 kW, 300 kW, 700 kW per vehicle). ERS always has policy support, and static charging powers match the main scenarios.

## Transport electrification, system cost, and greenhouse gas emissions

The simulations indicate that the system of EU road freight is undergoing a transition to a new state with both lower levelized costs and GHG emissions, under all conditions. This means that the best outcomes are those when electrification starts early, is quick, and includes as much traffic as possible. BETs can offer lower levelized transport costs than internal combustion engine trucks (ICETs) only when the charging infrastructure is sufficient in terms of both coverage, density, and charging power. Insufficient charging infrastructure makes electrification uneconomical. The terms transport cost and system cost are defined in "Methods," with the former capturing internalized costs that determine behavior, and the latter measuring socio-economic outcomes, including externalized costs.

In the simulated transport system, electrification is largely constrained by the rollout of charging infrastructure. If simulations are run with unconstrained access to charging infrastructure (not presented), BETs would achieve a levelized cost advantage already by 2025 over ICETs on most routes. This means that scenarios with fewer constraints on charging infrastructure placement generally also have more

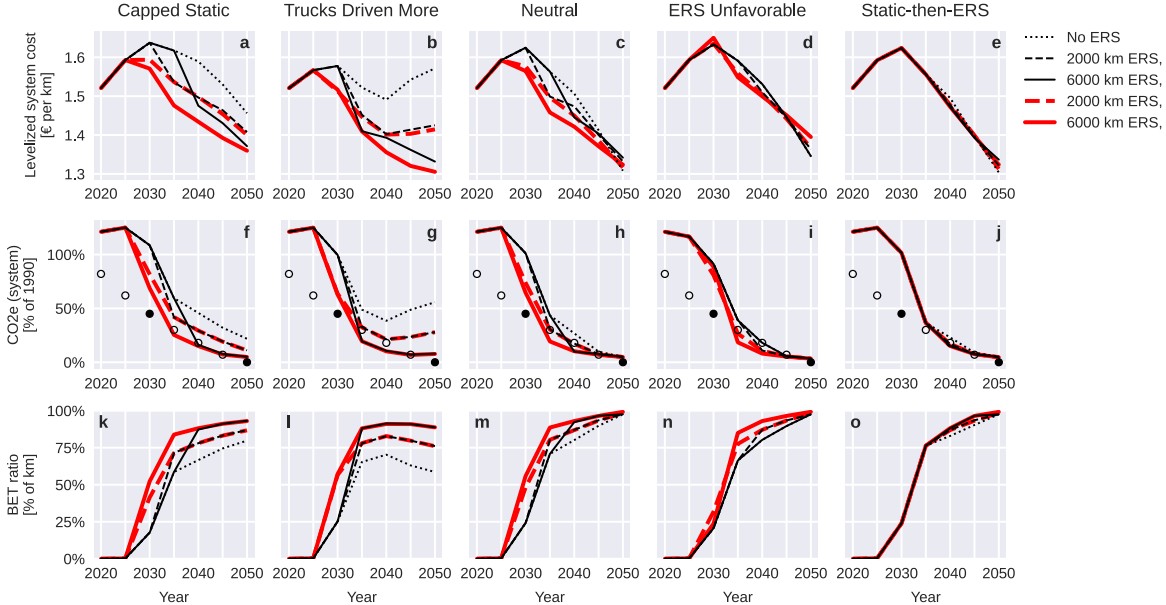

**Fig. 1 | Impact of including electric road systems (ERS) in the charging mix on levelized system cost, total greenhouse gas (GHG) emissions, and battery-electric truck (BET) share of traffic.** Investing in an ERS network is identified to be a no-regret strategy based on that costs (**a**–**e**) and GHG emissions (**f**–**j**) are reduced across all conditions as a result of the investment, versus equivalent scenarios without ERS. **k**–**o** ERS investment furthermore creates conditions for electrifying >80% of road freight (**k**–**o**) decades earlier and with greater probability than if only static charging is available. A price cap on ERS charging (policy support in the form of a conditional subsidy) reduces total system costs by attracting traffic to infrastructure with largely usage-independent costs. Scenarios shown are from Table 1, with four parameter conditions (columns), three ERS conditions (line pattern), and two policy conditions (line color). Filled circles indicate GHG reduction targets for the European Union set in the European Climate Law (2030 and 2050), with hollow circles indicating intermediary values interpolated by us.

favorable simulation outcomes. Building ERS in parallel with static charging makes it possible to reach infrastructure sufficiency earlier and along more routes, compared to if only static charging is available.

Figure 1 presents simulated shares of electrified transport work and resulting annual GHG emissions, for each 5-year period and for each of the 25 main scenarios. Scenarios without ERS are drawn as dotted black lines, and scenarios with policy-incentivized ERS use are drawn in red. Differences between black and red lines of the same pattern indicate policy impact. Transport electrification rates above 85% are achieved in all conditions with ERS, but not without. In Capped Static and Trucks Driven More, static charging alone leaves diesel as the cost-minimizing option for 20–40% of road freight even by 2050. With significant long-term benefits and failing transport electrification in the absence of ERS, full ERS network expansion is clearly warranted.

In the Neutral, ERS Unfavorable, and Static-then-ERS conditions, static charging alone can eventually enable full electrification of road freight. In these scenarios, the GHG impact of ERS is limited to the transition period (2030–2050), during which simulated cumulative emissions decrease by 6–43%. Expansive ERS networks (6000 km) accelerate the electrification of the last transport routes by approximately a decade. In the remaining two conditions (Capped Static and Trucks Driven More), ERS reduces 2030–2050 GHG emissions by up to 55–63%, and GHG emissions continue post-2050 without ERS.

In all scenarios, the greatest increase in BET ratio within a single 5-year period is around 50%, centered around 2030. From prior data analysis[7], we estimate that vehicles younger than 5 years represent approximately a third of European HDV transport work. Newer vehicles also contribute a greater share of transport work in North and West European countries than in South and East European countries. A transition nearly twice as fast as natural fleet turnover implies either increased vehicle scrappage and new sales, greater concentration of transport work on newer vehicles, or retrofits of ICETs to electric. It appears ERS would make all three pathways more feasible, by reducing 2030 BET purchase prices by approximately a third, by enabling a full transition in the Trucks Driven More condition, and by making retrofits more economical[7].

Figure 1 also presents distance-levelized total system costs. ERS with policy support contributes to short-term cost reductions in all scenarios where it is built early. In the two conditions where ERS is needed for full electrification, ERS also brings substantial long-term cost reductions (7–17% by 2050). Long-term cost impact in Neutral and Static-then-ERS is negligible. However, if society develops in the direction of ERS Unfavorable, with ERS budget overshoots and permanent increases in daytime electricity prices, ERS construction should likely be halted after the main long-distance corridors have been built, as further densification of the network would increase system costs.

The simulated policy for incentivizing ERS use has a negligible impact on the Fig. 1 indicators with the 2000 km network, and a positive impact in the Neutral and ERS-favorable conditions with the 6000 km network, in which both costs and GHG emissions are reduced. This supports prior research showing that full internalization of the ERS infrastructure cost is a socio-economically sub-optimal strategy[10,14].

Based on the spread of simulated outcomes, we conclude that (1) ERS eliminates the risk of a failed transition (due to factors incorporated in the simulation), (2) investment decisions about the first stages of an ERS network will have to be taken under uncertainty about the marginal socio-economic impact of the investment, (3) the worst-case outcome is neutral, (4) an early investment is always better than a late investment, and (5) minor subsidies of ERS user fees appear to reduce total system cost for expansive networks, including the cost of the subsidies.

## Demand for batteries and combustion engine fuels

Static charging alone achieves nearly complete electrification in the Neutral, ERS Unfavorable and Static-then-ERS scenarios, but only by 2050 when overnight (depot) charging is available for almost all routes (Fig. 1). The assumed supply of sustainable drop-in fuels[7] of primarily biological origin enables a complete fossil fuel phase-out at around 95% electrification (Fig. 2). ERS can contribute to reaching this target earlier and with greater probability. Without ERS, fossil fuel

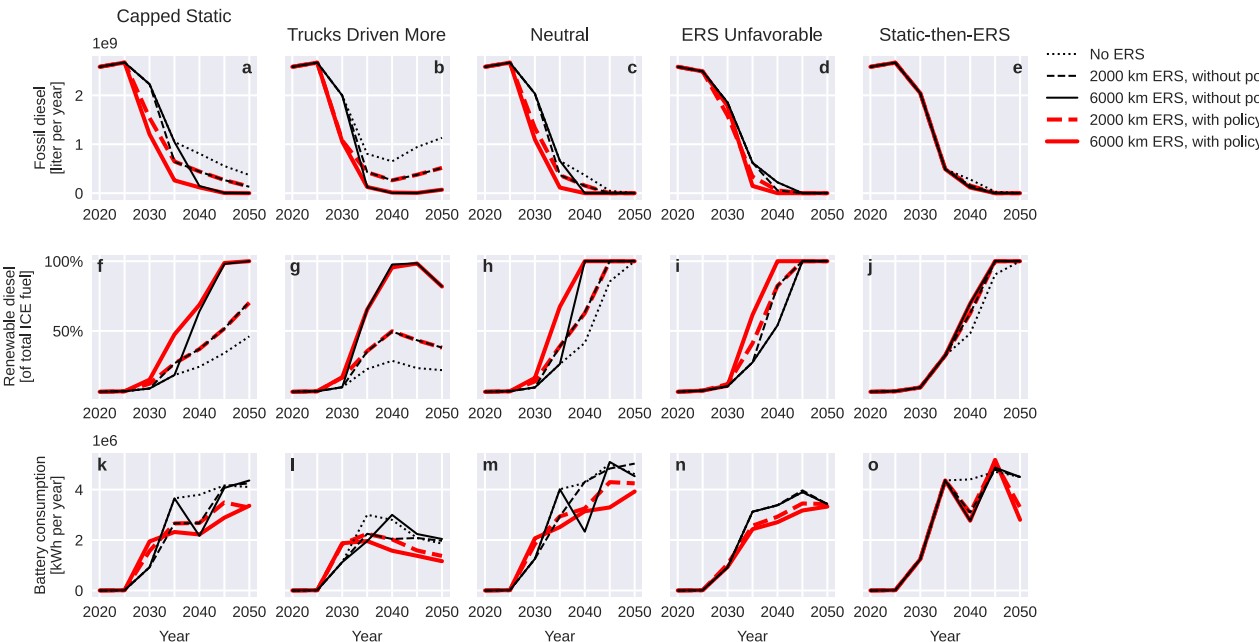

**Fig. 2 | Impact of including electric road systems (ERS) in the charging mix on demand for internal combustion engine (ICE) fuels (fossil and renewable) and battery cells for heavy trucks.** ERS makes a complete fossil fuel phase-out quicker and more likely (**a**–**e**). Total consumption of renewable ICE fuel was assumed to be supply-capped at 2020 levels with 2% increased annual supply, while the blend ratio (**f**–**j**) was adjusted to minimize the use of fossil fuels. High blend ratios for renewable fuels are made possible by low total ICE fuel demand. ERS also reduces total annual battery consumption (**k**–**o**, see main text for definition) by -25–40%, despite increased battery-electric truck uptake, with greater reductions for more expansive networks.

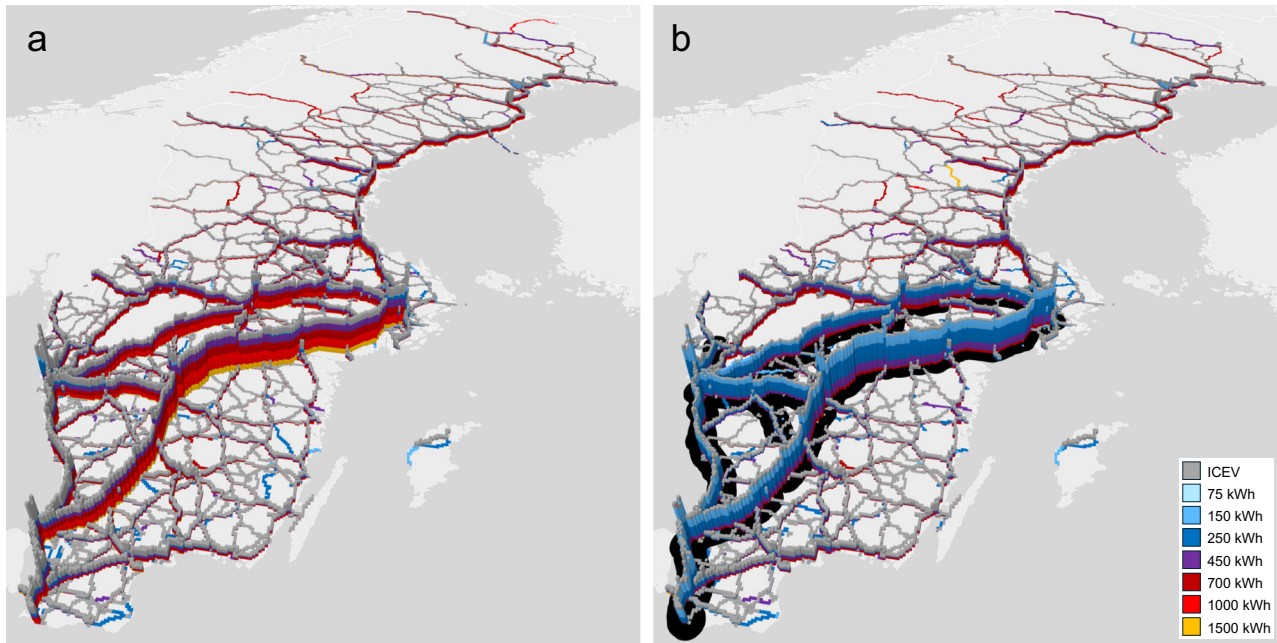

**Fig. 3 | Electric road system (ERS) impact on cost-minimizing battery capacity per truck.** Access to ERS reduces the cost-minimizing usable battery capacity per vehicle by ~70% in this scenario. The effect size depends on location along the road network, year, ERS network scope, static charging infrastructure, and global model parameters. The maps show the Neutral scenario in 2035, **a** without ERS; and **b** with policy-supported ERS on a 2000 km road network. Black indicates where ERS is available; other charging infrastructure is present but not shown. Bar height indicates traffic volume (vehicles per day), and color indicates the share of that traffic from vehicles equipped with different usable battery capacities (75–1500 kWh, light blue–orange). Country shapes from geoBoundaries[39], licensed under CC BY 4.0.

consumption persists beyond 2050 in the conditions Capped Static and Trucks Driven More, with biofuel supply limited to 25–50% of the total 2050 combustion fuel demand.

An unhindered transition to electric vehicles requires an adequate battery supply, in addition to charging infrastructure. The total rate of battery resource consumption (ignoring supply-demand price dynamics) is determined by the rate of BET adoption, battery pack capacity per vehicle, and the rate at which batteries are worn out through use. As the simulation model keeps track of transport work but not a stock of vehicles, we report battery demand as levelized consumption, i.e., the annual sum of battery ageing over all transport routes. A battery is considered consumed when it is retired from use in a truck, either because it has lost too much capacity or because the truck itself has reached its end of life. Second-use applications are possible, but are assumed not to be in other trucks.

Very rapid BET uptake is observed in all conditions, to higher levels in the presence of ERS. However, Fig. 2 shows how scenarios that include ERS simultaneously have the lowest total annual battery consumption. This is because the cost-minimizing truck configuration in the presence of ERS is to use a battery pack with ~70% lower capacity than if only static charging is available. Fig. 3 shows the distribution of cost-minimizing vehicle battery capacities along routes traversing different parts of the simulated road network. In the presence of ERS, the cost-minimizing usable battery capacity is reduced from ~450–1500 kWh per vehicle to 150–700 kWh per vehicle. The figure shows results for 2035. The simulation uses simplified physical volume and weight constraints and usable battery capacities of 700 kWh or more may not comply with EU regulations for maximum axle load or total tractor-trailer combination length.

Our findings about charging infrastructure influence on cost minimizing battery capacities extends prior research that identified that such a capacity reduction is logistically possible for both trucks and passenger cars[8,11,12,17,18], but which did not consider the economic

implications of (1) how capacity reduction increases the relative charging rate to increase battery ageing, (2) how dynamic charging reduces battery ageing by supplying power directly for propulsion and bypassing the battery pack, (3) how changed battery weight affects vehicle carrying capacity and energy consumption, (4) how effects on cost-optimal battery capacity influence levelized BET costs in relation to ICETs, and (5) how battery capacity caps the share of daily energy use covered by overnight charging.

In scenarios with policy-supported ERS networks (6000 km), annual battery consumption across all roads is reduced by ~40% versus only static charging. This appears to be a long-term effect, except in the ERS Unfavorable scenario, in which high daytime electricity costs likely warrant larger battery capacities to maximize overnight charging. Fig. 4 shows how greater reductions in battery consumption result from more expansive ERS networks (panel a, horizontal axis), higher ERS charging power (panel a, color), and thus greater shares of total energy delivered via ERS (panel b). Battery consumption is reduced both because reduced installed battery capacity reduces calendar ageing in absolute terms, and through reduced cycling when some energy can bypass the battery to be used directly for propulsion. Enabling battery capacity reductions significantly reduces levelized BET costs until ~2040 and raises the probability that ICETs can economically be converted to BETs[7].

Prior work found that more frequent but shorter rest stops, which are permitted under the EU driver rest legislation, also enable substantial reductions in battery capacity per truck[19]. We confirm this finding using MOSTACHI simulations (Supplementary Fig. 6). Furthermore, our simulations reveal that the impacts of ERS and increased rest stop frequency stack, yielding further increases in BET share and further reductions in GHG emissions. This suggests frequent stopping offers a complementary strategy that does not affect the qualitative conclusions of this research. It is however beyond the scope of this study to thoroughly analyze the combined strategies and frequent stopping may have logistical implications that are not simulated in MOSTACHI.

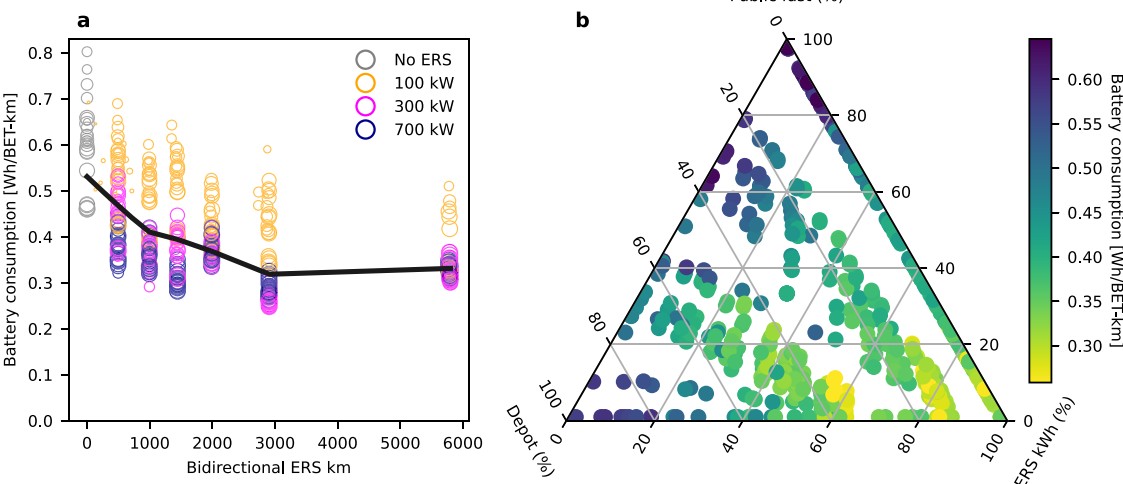

**Fig. 4 | Charging infrastructure impact on the consumption of batteries.** Consumption is defined here as the rate at which virgin gross battery capacity passes through the total battery-electric truck (BET) population, including both electric road system (ERS) and non-ERS adapted vehicles. Second-life applications are possible, but are assumed not to be in trucks. Each marker represents one scenario (see "Experimental design"). **a** Improved access to charging reduces distance-levelized battery consumption. Low-power ERS (orange) yields consumption rates down to ~30% below only static charging (gray), while higher-power ERS (magenta and purple) yields savings down to ~50%. ERS network expansion into regions with less traffic yields diminishing returns. Absolute capacity loss to calendar ageing is reduced with smaller batteries, while cycle ageing is reduced with lower charging rates and more energy directly to propulsion, bypassing the battery. The addition of ERS reduces mean annual consumption by up to 40% further. **b** The rate of battery consumption correlates strongly with the share of each method of delivery of electrical energy. Greater reliance on static charging, at destinations and rest stops (combined into Public fast) and depots, drives up distance-levelized battery consumption (blue), while dynamic charging reliance reduces battery consumption (yellow). System-wide battery consumption is shown in Fig. 2.

## Charging infrastructure demand

The shares of energy delivered via charging infrastructure at different locations depend on the composition of charging infrastructure in the transport system as well as on model parameters such as vehicle operating hours per day and daytime versus night-time electricity costs. Fig. 5 shows how total delivered electricity from each of the four types of charging infrastructure develops over time. All graphs begin at zero, as no charging infrastructure is offered in model year 2020. The graphs show aggregate demand across all sites of the same type and say nothing about between-site competition.

Scenarios with ERS but without policy support show erratic lines, indicating system instability and unpredictability in all conditions and for most charging types. The ERS network becomes a stranded asset by 2050 in the Neutral, ERS Unfavorable, and Static-then-ERS conditions. Investment in an ERS network while mandating that costs are fully internalized through user fees appears to be a risky proposition, which negatively affects also investors in vehicles and public static charging. ERS causes demand turbulence also with policy support in Static-then-ERS, as late ERS deployment prevents the market from reaching an equilibrium within the simulation timeframe. The simulation results suggest there are only downsides to first building one type of charging infrastructure and then replacing it with another type.

In the remaining scenarios, without ERS or ERS with price control, investment risk appears similar for all three types of daytime charging —demand decreases sharply post-2040 if the cost incentives for night-time charging are as strong as in ERS Unfavorable, and less sharply under the Neutral condition. Investors experience no significant demand volatility in Capped Static or Trucks Driven More.

Demand for overnight charging is largely unaffected by ERS investments. It grows steadily in all conditions except Trucks Driven More, as a single charge represents a smaller share of total energy when daily distances and operating hours increase.

Substitution effects between different types of charging infrastructure occur when new infrastructure attracts users from prior infrastructure. Synergies can also occur when new infrastructure increases total electrified traffic, adding new users throughout the system. These effects are quantified in Fig. 6 (see also "Experimental design") which shows the marginal change in total electrical energy delivered via each type of charging infrastructure caused by a marginal increase in availability of each other type of infrastructure.

The purpose of Fig. 6 is to better explain the risk that investors take when the future availability of competing infrastructure is uncertain. Results indicate that ERS in the charging mix reduces demand for fast charging at rest stops by 50–60%, at destinations by 35–50% and at depots by 20–30%, and that depot charging also replaces destination charging. These strong substitution effects indicate that an investor in ERS should strive for maximum transparency regarding the timeline and placement of the ERS network, to redirect investments in public static charging towards locations with long-term demand. Without transparency, investors in public fast charging infrastructure must compensate for the increased business risk through raised prices, which in turn reduces incentives to transition to electric vehicles.

## Monte Carlo sensitivity analysis

A Monte Carlo sensitivity analysis was run to understand if the simulation is particularly sensitive to changes in any of the input assumptions. Five hundred randomized conditions were simulated for three model years (2040, 2045, 2050) with and without ERS in the charging mix (3000 simulated scenarios), with each 5-year period in each condition (n = 1500) represented by one marker in Fig. 7. The figure shows how the benefits of having ERS in the system emerge under conditions when static charging would not have been enough to fully electrify road freight (orange and cyan markers). Under conditions when the transition could have succeeded anyway (purple markers), the system cost is similar with or without ERS.

The sensitivity analysis reveals that decreasing vehicle utilization, a road network with insufficient traffic, a too small ERS network, too low ERS power, or very high ERS cost would each reduce the marginal benefit of an ERS network. Conversely, if vehicle utilization increases, ERS is necessary for the transition to succeed. There is a weak trend that the marginal benefit of ERS decreases by 2050 versus 2040.

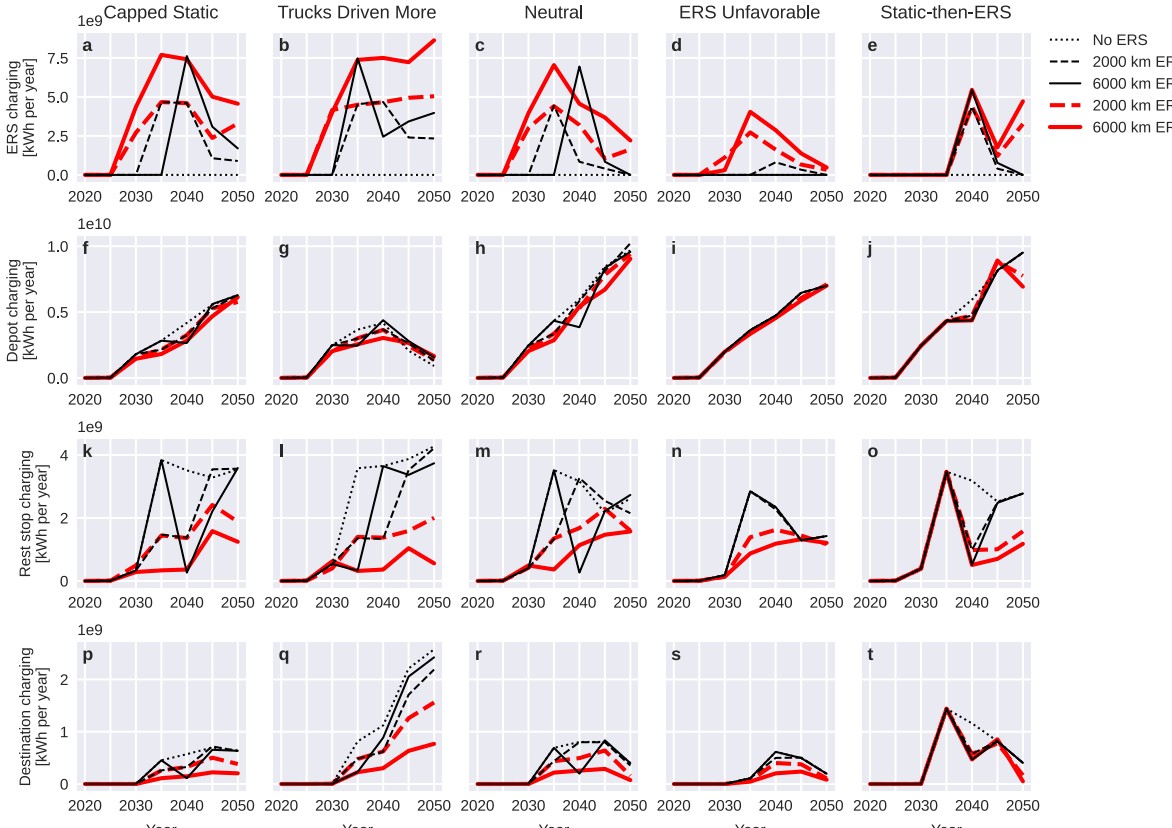

**Fig. 5 | Impact of including electric road systems (ERS) in the charging mix on demand for charging from different types of charging infrastructure, measured as total delivered energy.** Scenarios shown are from Table 1, with four parameter conditions (columns), three ERS conditions (line pattern), and two policy conditions (line color). **a–e** Policy support greatly increases ERS charging demand and thus reduces investment risk in ERS networks under assumed traffic intensities (2000–10,000 bidirectional truck passages per day). See further discussion about the ERS Unfavorable condition in the main text. **f–j** Demand for overnight charging at truck depots is unaffected by ERS availability, as increased

BET uptake (Fig. 1k–o) counteracts decreased charging demand per truck (capped by battery capacity, Fig. 3). **k–t** Demand for public fast charging at rest stops and logistics hubs is sensitive to competition by ERS (see also Fig. 6), and policy support for ERS makes demand less volatile than with unsupported ERS. All four charging types have a charging demand by 2050 that is at a level substantially below an earlier peak under at least one condition. While this study focuses on de-risking ERS investments, our simulation results indicate a need for de-risking strategies also for static charging.

However, MOSTACHI is at present only designed to study the transition period, and model improvements would be needed to make accurate statements about the transport system beyond 2050.

## Discussion

Prior work has indicated that while dynamic charging via ERS of heavy-duty vehicles would be cost-efficient today, future technoeconomic developments are possible that may cause eventual abandonment of the infrastructure, due to negative feedback loops. Our simulations take additional dependencies into account and confirm this dynamic. Abandonment is the flipside of strong economies of scale and risks occurring if a sufficiently large minority of users can reduce their own total operating costs by switching to another charging solution. A shrinking user base reduces revenue for the ERS operator, who will only incentivize further defection if they raise prices.

ERS construction followed by abandonment is highly undesirable for the transport system, as such turbulence raises risks for investors in both charging infrastructure, power grids, and vehicles. Furthermore, our simulations show that although the first defectors who trigger the breakdown will most likely switch to other charging strategies, the last defectors are more likely to return to fossil fuel use.

The primary contribution of this research is that we propose and show that the negative feedback loop can be broken and ERS investments secured by policy instruments targeted at preventing these

initial users from defecting. We test a policy implementation where the ERS user fee is capped near the average price for public fast charging at rest stops, i.e., introducing subsidies if costs surpass revenue. This implementation is motivated by prior work that showed full internalization of ERS costs through user fees yields sub-optimal socioeconomic outcomes[10,14].

Our simulations reveal that when accompanied by such a policy, the addition of ERS to the charging mix ensures that fossil fuels are phased out from road freight before 2050 under all evaluated conditions, while simultaneously minimizing total system cost, median transport cost, and the spread of levelized operating costs for competing transport operators. When ERS is absent from the charging mix, we find that deep decarbonization of road freight is possible but far from guaranteed and that a full transition may be resisted by the market. With ERS, the same rate of GHG emissions reductions can be expected from road freight as from other EU sectors. Importantly, we find no socio-economic benefits of not investing in ERS or of delaying an investment decision. We do however, find harmful effects of late ERS deployment and of ERS deployment without accompanying policy support in high-risk regions, such as road networks with sparser traffic. Furthermore, we find that decarbonization of road freight is highly likely to become constrained by the natural rate of fleet renewal, and that ERS can facilitate both replacement with new vehicles, retrofits to electric propulsion, and allocating more transport work to newer vehicles.

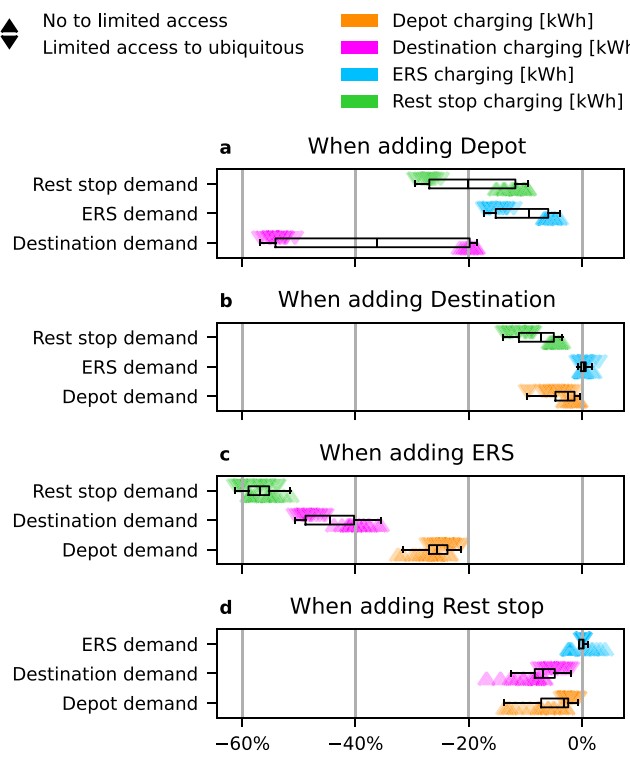

**Fig. 6 | Substitution and synergy effects.** When additional charging infrastructure is introduced into the system, this can compete with or support existing infrastructure. Each marker shows the change in total delivered energy from one step of increase in the availability of another type of infrastructure ($n = 54$ samples per type). See "Experimental design" for details. Adding ERS to the charging mix (subplot **c**) reduces demand for fast charging at rest stops by 50–60% (green), at destinations by 35–50% (magenta), and slower charging at depots by 20–30% (orange). Depot charging is a direct substitute for destination charging (**a**, magenta), though this effect may be overestimated by the representation of traffic as routes between origin-destination pairs. Demand for ERS charging is insensitive to static charging availability (subplots **a**, **b** & **d**, light blue). Boxplots extend from the first quartile to the third quartile of the data, with a line at the median. The whiskers extend from the box to the farthest data point lying within 1.5× the inter-quartile range from the box. Effect sizes vary with global model parameter assumptions, local transport patterns, and local road network conditions.

Our simulation model does not constrain vehicle configurations by axle layout or legal length. In practice, cost-minimizing battery capacities identified in scenarios without ERS—typically 700 kWh and above—may not be physically compatible with axle load limits of current roads and with the EU's legal length limits for tractor-trailer combinations. Raising length or axle limits is possible and could ease electrification while negatively affecting turning radii, safety, and infrastructure compatibility. Capping battery pack capacity around 700 kWh would significantly reduce simulated BET uptake in scenarios without ERS and would make ERS investments less risky in scenarios without other supportive policies. By making logistics viable with smaller battery packs, ERS-equipped trucks would be more likely to stay within historical weight and length limits, avoiding such trade-offs. Even if longer and heavier vehicles are permitted, smaller battery packs leave more capacity for cargo.

Based on these findings, we conclude that early large-scale deployment of ERS is a risk-minimizing no-regret strategy for Europe, provided it is accompanied by a state-guaranteed dynamic price cap or equivalent policy instrument. Under conditions where such a dynamic price cap would translate into a subsidy, the policy would be correcting for market failure to ensure optimal socio-economic outcomes.

To maximize the probability of positive outcomes and minimize the risk of negative outcomes, a European decision to invest in a pan-European ERS network must be taken very soon. It will not be possible to say precisely what the marginal return on the investment will be. However, policy support can ensure a non-negative outcome under all evaluated scenarios. Prior research has identified substantial differences in willingness to pay for ERS charging between different user groups and different parts of the road network[14], suggesting that differentiated pricing should be explored to minimize reliance on subsidies and further increase ERS revenue without reducing utilization rates. ERS utilization by even a minority of light-duty traffic would also contribute strongly to ensuring a low levelized infrastructure cost, particularly near major urban centers. Our experiments used the Swedish road network layout with traffic densities artificially raised to match an average European country. As ERS economics are sensitive to traffic density[9], outcomes will be more favorable than shown here on the TEN-T corridors with densest traffic. Conversely, on road networks with sparser traffic, ERS may not be a no-regret investment—in particular as utilization in peripheral European countries may depend heavily on ERS availability in more populous neighboring countries[20].

We recommend staged construction, beginning with the no-regret main transport corridors within the TEN-T network (e.g., Manchester–London–Antwerp–Paris; Amsterdam–Rotterdam–Antwerp–Essen–Hanover–Berlin; Warsaw–Kraków–Wrocław–Warsaw; and Bologna–Milan–Venice). In 2019, Europe (excluding France) had approximately 17,000 km of motorways with daily HDV traffic above 6000[21], which we believe is a reasonable scope for the first stage of a European deployment. While smaller-scope investments may be economically viable, only a rapid international scale-up would contribute substantially to transport decarbonization. By the early 2030s, when the time comes to make final decisions about low-regret second-stage investments to connect and expand these corridors, more information will be available regarding key decision criteria such as construction costs, electricity price developments, progress on static charging infrastructure, and ERS impact on logistics. Such a core network would facilitate organic regional expansion by the late 2030s and early 2040s, including into and out of cities. Our simulations indicate that a well-planned expansive European ERS network would deliver up to 70% of all energy to BETs, while less expansive networks would have strong but localized effects. While our focus is on ERS impact analysis, our results indicate that infrastructure for overnight charging of parked trucks is also of critical importance for a rapid transition to electrified road freight.

## Methods
### Summary
We introduce MOSTACHI, an agent-based simulation tool designed to study interaction effects in time and space between logistics patterns, competing charging infrastructure, and cost-minimizing vehicle operators. MOSTACHI employs agent-based optimization techniques to minimize the levelized cost of transporting fixed amounts of goods along overlapping unidirectional transport routes. Through comparison of simulated scenarios, MOSTACHI supports quantitative estimation of how changes in technoeconomic conditions or changes in policy instruments influence emergent behavior resulting from system dynamics in a national or larger road transport system.

The spatial resolution of the model is defined by the input data. Our experiments use road segments approximately hundreds to thousands of meters in length. The temporal resolution of the model is mixed, with driving simulated per road segment along routes, iteratively in 5-year steps with changing infrastructure and parameter conditions. Four weight classes of trucks are currently simulated. Trucks are either equipped with internal combustion engines (ICETs) or battery electric (BETs)—see "Model limitations" for discussion about hybrid and fuel cell electric trucks.

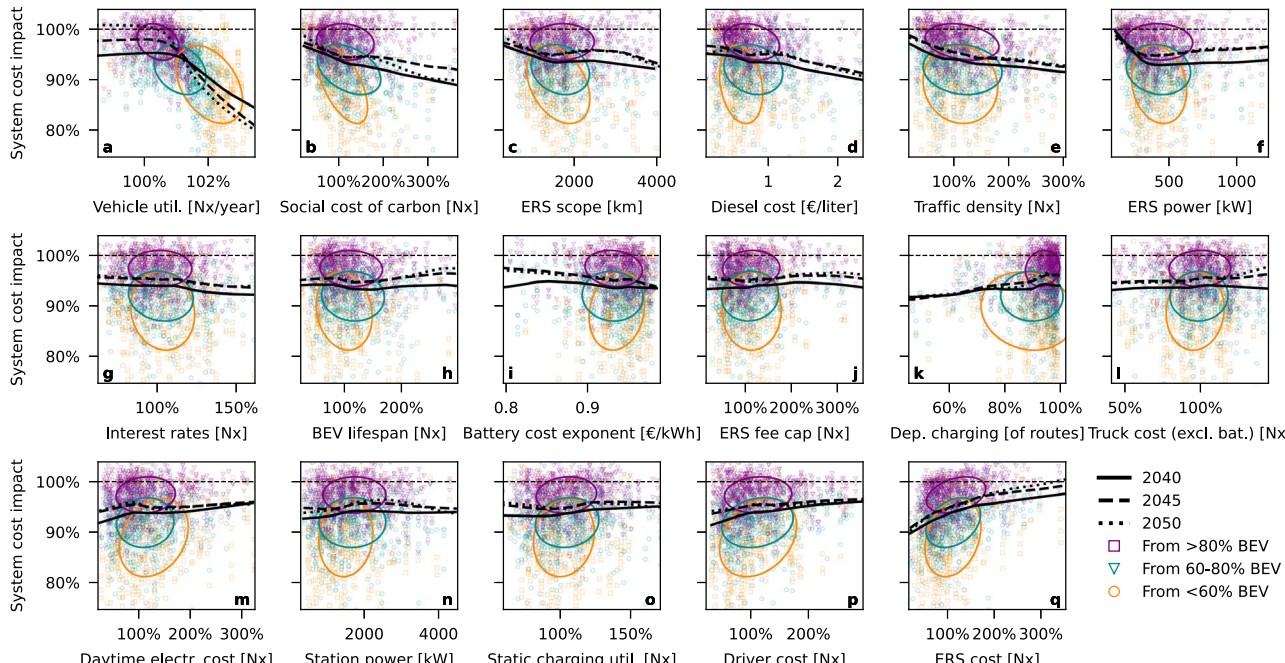

**Fig. 7 | Monte Carlo sensitivity analysis.** We conclude that the socio-economic value of an electric road system (ERS) investment decreases with reduced per-vehicle utilization (**a**), a low valuation of fossil carbon (**b**), limited ERS scope (**c**), low traffic density (**e**), low maximum ERS power per vehicle (**f**), and high ERS deployment cost (**q**). We also conclude that the addition of ERS substantially reduces system costs in scenarios where static charging alone fails to electrify all road freight. At worst, in scenarios where full decarbonization would have been feasible with only static charging, ERS does not substantially raise system costs. Static charging alone does not allow the electrification of vehicles with high utilization. Each subplot (**a–q**) shows how sensitive the model is with respect to each of 17 input variables that the authors assumed could significantly impact the simulation, based on our understanding of the world and the model. Horizontal axes indicate

the sampled value of the input parameter. Each marker represents change between two runs with identical randomized input parameters ($n = 1500$, markers repeat between subplots), differing only in the inclusion of ERS in the charging mix. The vertical axis indicates the relative change in total annualized system cost with ERS compared with no ERS. Marker color (purple, cyan, or orange) indicates how much traffic was electrified in that scenario with only static charging. Ellipses represent the 2σ covariance region of the bivariate normal distribution fitted to each group (covering ~95% of the data points), illustrating the principal axes and spread. Black solid, dashed, and dotted curves indicate a nonparametric lowess model (locally weighted linear regression) fitted to the samples from model years 2040, 2045, and 2050.

Simulated freight takes place along routes in an origin-destination matrix. Charging infrastructure can be placed at the beginning of routes ("depots"), at segments along a discretized road network (ERS and at "rest stops"), and at the end of routes ("destinations"). Charging installations are sized and priced in a supply-demand feedback loop. Charging infrastructure expands iteratively in 5-year increments, and competition arises in space and time. Vehicle traction batteries are included in the cost to be minimized. Battery lifecycle costs are determined through simulated charging behaviors, also subject to optimization.

The computational flow is conceptually illustrated in Fig. 8. Inputs to the model are ~200 global parameter estimates for each of seven simulated 5-year periods (2020–2054), a weighted origin-destination (OD) matrix of annual tonnage and trips per vehicle class and route, and a scenario to evaluate. All parameter values are listed in Supplementary Tables 1–13. Scenarios define the maximum availability of charging infrastructure at each type of location per 5-year period and may include alterations of global model parameters.

For each 5-year period and each combination of vehicle class and route (origin-destination pair), the cost-minimizing combination of powertrain (ICET or BET), battery capacity, and charging strategy is identified. Total per-site charging demand from all routes is summed and used to estimate the levelized cost of charging per infrastructure site (subject to scale benefits). This cost minimization repeats iteratively until convergence. The resulting charging infrastructure is retained, and the simulation continues for the next 5-year period. Costs of charging are set for the charging operator

to recoup costs within an expected discounting period, including profit margin. If the charging demand at a site decreases below the inherited capacity, this results in increased user fees to cover site costs. Within this discounting period, outcompeted charging infrastructure is retained to estimate the total system cost. Charging infrastructure can be downsized or decommissioned past its discounting period.

We model road freight in a European context, but as we build on prior work from Sweden, we use calibrated data[22] of road freight inside, entering or leaving Sweden, from the nationally well-established Samgods model[23]. The data represent an origin-destination (OD) matrix of total annual goods tonnage transported, and total number of trips in 2017, between ~200,000 pairs of regions inside and outside of Sweden, for four truck weight classes 16–60 tons. Equivalent OD matrix data exist for other parts of the world, e.g., between the European NUTS3 regions[15], making the methodology generalizable. Start and end points within regions for OD pairs are sampled from a density distribution of areas likely to be involved in road freight supply chains, with routes between start and end points identified using OpenStreetMap and the Open Source Routing Machine.

Generalization of the simulation from Sweden to a European context is made possible by adjusting model parameters from the Swedish context (electricity prices and GHG emissions, GHG valuation, biofuel supply cap, etc.). We strive to exclude the influence of any subsidies, except for fossil GHG emissions, which are not taxed at the full social cost of carbon.

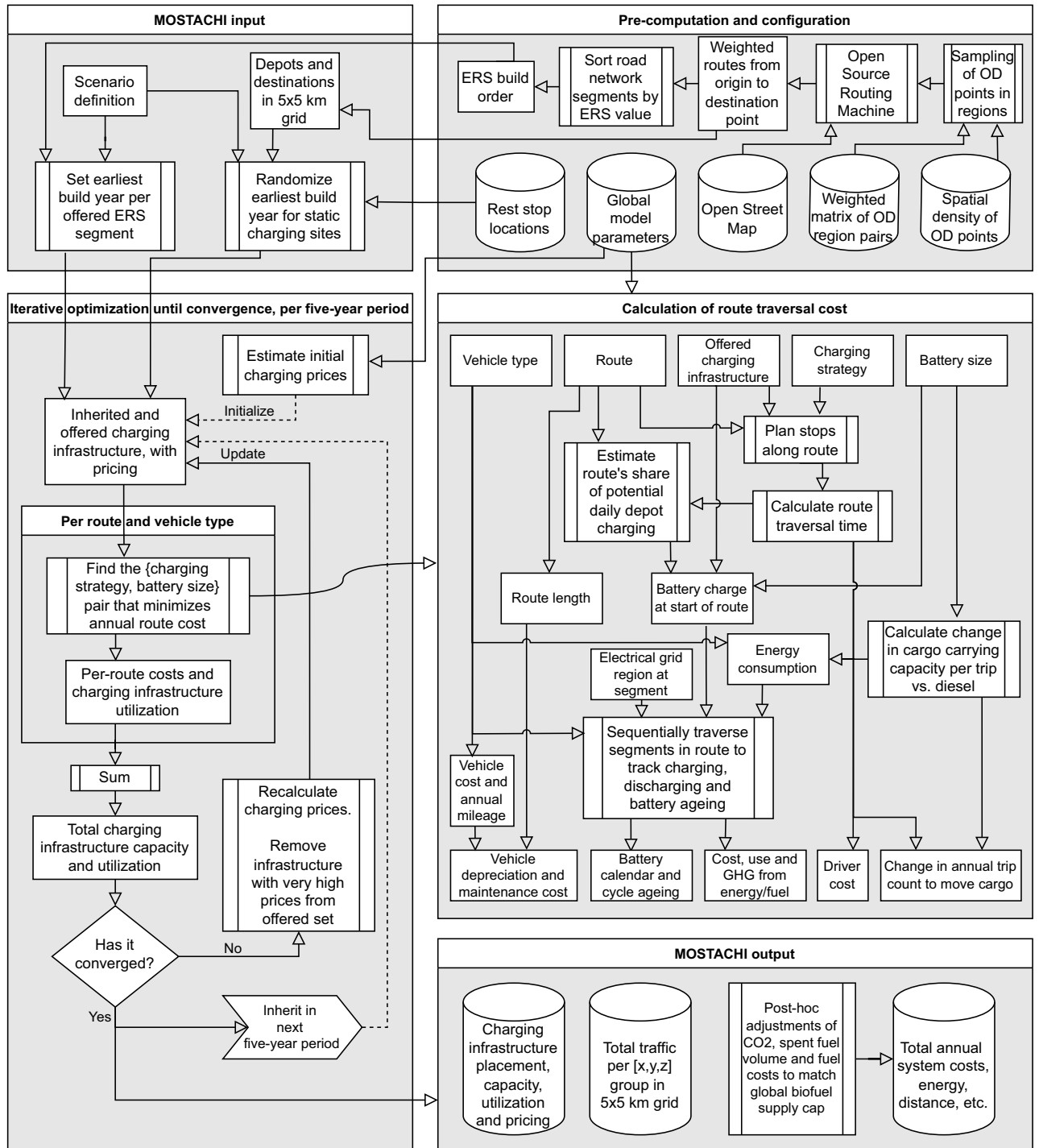

**Fig. 8 | MOSTACHI flow chart.** The figure depicts the computational flow in the simulation tool used for experiments in this study. See the main text for details and "Model calculations" for explanations of individual steps.

## Study design limitations and implications

The simulations in this work do not constrain the rate at which the vehicle stock is replaced. Parallel work has highlighted how newly produced BETs are primarily going to contribute to the decarbonization of traffic in first-use markets, while markets dominated by second- and third-hand vehicles will lag in BET uptake[7]. Parallel decarbonization of all markets would require either sustainable drop-in fuels or e-retrofits, where combustion powertrains are stripped out and replaced with battery-electric powertrains. Biofuels are in too short supply while e-fuels are both scarce and expensive[7], and substantial barriers to

e-retrofits have been identified in terms of vehicle conversion cost, poor vehicle range due to weight-constrained batteries, and limited EU battery supply[7,24]. The battery reductions at both the vehicle and fleet levels enabled by ERS would make retrofits more feasible, but the relationship is not modeled explicitly in this study.

Experiments are set up such that investments in ERS are in addition to other investments in infrastructure for static charging. This implicitly assumes multiple independent investors, as opposed to trying to find an optimal way of spending a fixed total infrastructure budget. The scope of this research is limited to understanding the

impact that ERS would have on the transport system if implemented at scale. This assumes reliable and standardized technology, not too far from assumed costs. To date, no ERS technology has been deployed commercially at scale. It is not within the scope of this research to address unanswered questions regarding road durability impact, long-term maintenance costs, cybersecurity, supply chain capacity, and electromagnetic compatibility[25,26]. However, deployment is held back more by political and bureaucratic bottlenecks than technical readiness[7], and coordinated effort is needed to reach agreement on international standards and to integrate ERS into regulatory frameworks, such as the EU's Alternative Fuels Infrastructure Regulation[27,28]. Policy research is needed to understand how to best incentivize ERS use with minimal adverse effects, and MOSTACHI can be used to validate expectations of policy impact.

The study is also limited to analyzing heavy-duty road freight. Road-bound ERS technologies generally support charging of both light- and heavy-duty vehicles. Prior research suggests the environmental benefits of ERS are significantly increased if light-duty vehicles also use the ERS network, as the main source of fossil carbon emissions from a future electrified road transport system is resource extraction and manufacturing of batteries for passenger cars[7]. The economic incentives for ERS use by a subset of light-duty vehicle traffic appear to be sufficient to warrant consideration of their logistic needs in the planning of a future ERS network[7,17,18].

Our analysis of the Swedish road network topology and its transport patterns may introduce biases that are difficult to predict. Key model parameters have been adjusted where Sweden notably diverges from the EU average: biofuel supply cap (reduced from 35% of total combustion fuel consumption in 2020 to 5%), electricity price (30% lower in Sweden) and GHG intensity (65% lower in Sweden in 2020), and $CO_2$ valuation (reduced from 700 €/ton to 200 €/ton).

By international comparison, Sweden exhibits relatively sparse traffic, and most border crossings are concentrated on relatively few bridges and ferry lines. Annual average daily traffic (AADT) by heavy vehicles on the core Swedish road network is 1000–5000, which was multiplied by 2.5 in the simulations to approximate average EU conditions. Main roads in central Europe are in the 6000–18,000 range[21]. Denser traffic improves ERS economics, reducing the risk of ERS abandonment. Inclusion of light-duty traffic is thus likely to further reduce socio-economic risk. While electricity prices are significantly higher during the day than at night in Sweden and our experiments, the pattern may be reversed in regions with a high supply of solar power, which would further favor ERS. Limitations in the employed simulation model and input data are discussed in subsequent sections.

## Model design and architecture

MOSTACHI's simulation logic is conceptually very simple. Initially, it requires defining (a) global model parameters that represent the national context, such as expected electricity prices and interest rates for the studied period, and (b) one or multiple scenarios that specify the upper limit of the scope of charging infrastructure expansion. Subsequently, the model iterates over the desired timeframe (currently 2020–2050 in 5-year increments) and for each period it performs independent cost-minimization for overlapping transport routes. For each route, it is decided whether a diesel or battery-electric powertrain would offer the lowest total transport cost, and for electric powertrains, the optimum combination of battery size and charging locations. The model then sums the demand for charging (peak electrical power) at all locations considered in that time period, adjusts the estimated pricing of charging at each location, and repeats the calculation with the new price information. When charging patterns have stabilized, the charging infrastructure is fixed (built) for that period. Built infrastructure is inherited from one period to the next, but vehicles are not, as it is assumed that vehicles can be resold to operators in other locations. Charging infrastructure is permitted to be downsized or decommissioned beyond the economic lifetime used to estimate user charges.

MOSTACHI models four categories of charging infrastructure for electric vehicles: dynamic charging (electric roads), and three forms of static charging (truck depot charging, route destination charging, and charging at major driver rest stops). The model accounts for economies of scale for all types of charging infrastructure, i.e., that larger facilities yield a lower cost per energy unit given the same rate of utilization. The extent of such scale benefits varies, being minimal for depot and destination charging, modest for rest stop charging, and substantial for ERS.

Simulation of one 5-year period takes place as follows. (1) Any already built charging infrastructure is inherited from the previous simulation period. (2) According to limits imposed by the scenario specification, charging infrastructure at all permitted locations is treated as if it were built. (3) The price of charging at each location is estimated using a heuristic. (4) For each transport route and vehicle class, the route is traversed one road segment at a time to identify which combination of powertrain (ICET or BET), battery storage capacity, and charging strategy would result in the lowest total annual cost of transporting all goods along that route. (5) All charging for each charging infrastructure location is summed, resulting in an aggregate demand, which can be above or below any inherited capacity. (6) The cost of charging at each location is updated based on estimated utilization and any inherited overcapacity. Once past the depreciation period, installed capacity can be reduced to match demand. (7) If the cost of charging at a site is found to be more than five times the global average for that charging infrastructure type, that site is removed as a candidate for infrastructure construction in this period. (8) The simulation is repeated until the total energy delivered per type of charging infrastructure has stabilized. The simulation flow is represented visually in Fig. 8.

The iterative optimization until convergence within time periods, of vehicle configurations, charging strategies and charging infrastructure expansion, is a computational way of handling that (a) the cost of charging at a location depends on how much charging takes place there, (b) where charging takes place and how much energy is purchased during a charge session depends on the vehicle battery's total storage capacity and current state of charge (SoC), and (c) both optimal battery capacity and current SoC depend on available charging infrastructure and pricing per location.

MOSTACHI embodies a free-market model, representing independent decision-making processes by vehicle owners regarding vehicle purchases and charging locations. It also allows for competing charging infrastructure established in different locations by multiple operators. While more resource-efficient system configurations might exist than those resulting from this simulation, it is unclear what incentives could steer an uncoordinated market economy towards such a long-term global optimum.

## Model capabilities

MOSTACHI captures several real-world system dynamics. The total amount of traffic and the energy consumption per vehicle change for each simulation period. All vehicle operators are assumed to act to minimize their own costs, and electrification only occurs when and where switching to a BET lowers the cost compared to using an ICET. The break-even point between BET and ICET operation is an emergent property determined by economic parameters, the state of technological development, operating patterns, charging infrastructure along the route, and opportunities to share the infrastructure cost with other routes.

Vehicles traversing different routes share public infrastructure, and routes that share an origin or destination also share depot and destination charging infrastructure. Charging stops that are not aligned with legally mandated driver rest periods increase transport

costs. Vehicles in each of the four different weight classes have different energy consumption profiles and gain different ranges from the same charging power. Battery capacity affects vehicle weight, which in turn influences energy consumption and carrying capacity, and not all battery capacities are available to all weight classes. Battery output power scales with battery capacity, and minimum output power requirements impose a lower bound on battery size.

At each simulation step along a route, the maximum charging power is constrained by the vehicle's current SoC (which limits the energy that can be received), the battery size (which limits maximum power), and the power rating of the charging infrastructure. Increasing the maximum power of charging infrastructure can shorten charging sessions by transferring more energy per unit of time, but also increases battery wear and infrastructure costs. Maximum power is further constrained by the battery's own ability to receive high charging rates, which improves with larger battery packs.

In the case of ERS, very high charging power can rapidly fill the battery, reducing subsequent power demand on downstream ERS segments. Energy charged via ERS can extend battery range outside the ERS network, but routes near but not on ERS stretches gain no benefit. All battery cycling contributes to battery ageing, which increases with higher C-rates (power relative to battery capacity). Because ERS can deliver power directly to propulsion, bypassing the battery, it reduces cycling-related degradation.

Powertrain selection, battery capacity, and where to charge are jointly optimized for each route to minimize total freight cost. Rapid improvements in battery technology and declining levelized battery costs influence the entire system, particularly by reducing the value of infrastructure that enables battery downsizing. Installed battery capacity also affects cargo carrying capacity, which in turn affects the number of trips required to move the same amount of cargo, influencing both driver and vehicle costs.

Only charging infrastructure with a sufficient user base to be profitable is actually built. New charging infrastructure can both complement and compete with existing infrastructure on intersecting routes, with ripple effects throughout the network. All types of infrastructure require time to build out, and the availability of options increases over time. Infrastructure that becomes available later may outcompete earlier installations. Shifting economic conditions and technical characteristics can alter charging demand over the infrastructure's lifetime.

ERS infrastructure is never built unless it attracts a critical mass of users. If that critical mass is lost later, the costs borne by remaining users rise. An early decision to build ERS entails economic risk due to the current lack of standardization, but delaying the decision increases the risk that earlier investments in static charging infrastructure will become stranded assets.

Expansion of a small ERS network increases utilization of already-built ERS segments because ERS-adaptation becomes a cost-saving decision on additional routes. However, there is a threshold beyond which further expansion reduces utilization, as the marginal benefit of additional ERS drops below its marginal cost. This threshold depends on several factors, including ERS construction cost, traffic density, traffic patterns, the cost and availability of alternative charging infrastructure, and battery costs. Road networks with greater traffic density can thus sustain ERS networks of greater scope.

Finally, usage fees are updated in 5-year intervals to recover the levelized infrastructure cost at current utilization levels. ERS fees are set globally, while static charging is priced per site. The model also dynamically calculates the blend ratio of fossil and renewable diesel based on total fuel demand and a predefined cap on renewable fuel supply.

## Model limitations

Several system dynamics are known to act on the real world but are not captured in MOSTACHI. These should be considered when interpreting the results. First, the availability of charging infrastructure outside the simulated region (Sweden, in this study) can influence demand for charging within the region. In the current model, route segments that fall outside Sweden are assumed to have access to charging infrastructure at equivalent placement and density as those within the simulated region.

Second, the model does not simulate a mixed-age fleet. In reality, vehicles manufactured in different years with different technologies operate simultaneously, but for computational simplicity, the model assumes full vehicle replacement every 5 years. Moreover, there may be upper limits on how quickly the fleet can be replaced, though historical European replacement rates—driven by tightened emissions regulations—may not predict future behavior. MOSTACHI handles replacement rate constraints only indirectly, via limits on charging infrastructure rollout specified in the scenario definitions.

Third, while vehicle and battery lifespans may in practice be coupled (particularly if batteries are structurally integrated with the vehicle frame), the model treats them as independent. This is based on the rationale that if the lifespans diverge substantially, batteries should eventually become a replaceable or reusable component.

The model also does not account for the price elasticity of transport demand. Changes in transport cost may alter traffic volumes across the network, which could affect infrastructure utilization, road congestion, travel times, and route choices, and may ultimately drive changes in the road network itself. Additionally, the route choice for each origin-destination pair is fixed in the simulation due to computational constraints. While charging availability may influence route selection in reality, the authors find it unlikely that such rerouting would significantly affect charging infrastructure economics, particularly since major roads already attract most traffic and infrastructure.

Vehicles that in reality operate on multiple routes must meet all constraints across those routes. In the model, optimization is performed independently per route, though routes remain interconnected via shared infrastructure use. Moreover, the model does not simulate dynamic daily operating behavior in response to electricity price fluctuations or charging congestion. Time-of-day adaptations could improve infrastructure utilization and reduce charging costs. Although the study assumes that daytime electricity prices are higher than night-time prices—thus favoring night charging—it does not model dynamic temporal behavior.

Another limitation is that aggregate demand for batteries, fuels, or electricity is not assumed to influence prices on the same scale, even though this could occur at broader geographic scales. Furthermore, a shift from high OpEx to high CapEx and access to dynamic charging could incentivize increased vehicle utilization, while high reliance on static charging, especially at night, could incentivize downtime. The model does not represent this dependency.

The model does not capture how the adoption of autonomous trucks may alter future charging demand, nor how the composition of charging infrastructure may affect how easily autonomous trucks can be introduced.

The simulations do not differentiate between loaded and unloaded trips, due to a lack of route-specific data. This simplification limits the model's precision if used for exact planning of charging infrastructure placement and quantity, although it is sufficient for studying interaction effects between infrastructure types.

Regarding vehicle technology, the model simulates only ICETs and BETs. Plug-in diesel-electric hybrids, while once considered a transitional option—particularly with ERS—are excluded based on discussions with truck manufacturers, who indicate they are no longer investing in combustion engine research and development. The authors hypothesize that adding hybrids would marginally increase ERS utilization and reduce static charging use, reinforcing the findings of this study. The omission of hybrids includes BETs with temporarily installed range-extending auxiliary power units.

Other propulsion types, such as hydrogen fuel cells and hydrogen combustion engines, are not included. This is due to their consistently higher projected total cost of ownership compared to ICETs and BETs across all simulated years[7,29–33]. Since MOSTACHI only selects the lowest-levelized-cost vehicle type per route and year, including options that are never cheapest would only increase computational burden. Methane-fueled combustion trucks were also excluded, due to their low market penetration and similar emissions profile to diesel.

Finally, the model does not incorporate current legal constraints on axle configuration or vehicle length. Battery packs above roughly 600 kWh usable (or more in the future, with improved energy density) increase axle load compared with diesel trucks, particularly in 4 × 2 configurations. To stay within axle load limits, a shift to 6 × 2 or 6 × 4 layouts may be required, which adds a third axle and reduces available space for batteries unless vehicle length increases. Simulated trucks with very high battery capacity (up to 1500 kWh) may therefore exceed length limits for articulated tractor-trailer combinations unless paired with shorter trailers or granted legal exceptions.

Other physical realities not modeled include mountainous terrain, ambient temperature, weather conditions, and traffic congestion. These may affect energy demand and ERS utility, though their exact impact is uncertain. Optimal placement of public static charging could be affected, for instance, by time-varying driving speeds or by weather-dependent energy consumption rates.

## Terms and definitions
Here follows a list of explanations of terms used in the "Methods" section and main text. Definitions of costs and model inputs are found in separate sections.

**Simulation model, input data, scenario, experiment.** The logic represented by the MOSTACHI source code is what we refer to as the simulation model. Input data used by the model are numeric constants (parameters) and transport routes, and pre-processed intermediary representations of these data. The input data represent the world to be analyzed. A MOSTACHI scenario is a combination of (1) maximum charging infrastructure availability per simulation period (including size of the ERS network); (2) ERS network electrification rate (gap ratio); (3) maximum charging power per vehicle per type of infrastructure; (4) available battery capacity options per vehicle type; (5) available charging strategies; and (6) deviations from the default parameter values.

Experiments are conducted by computing several scenarios and comparing differences in simulation results.

**Road segment, depot, destination, rest stop, charging station.** A road segment is the graph theoretic representation of a short section of road (edges) between two road network intersections (nodes). Truck depots, destinations and rest stops are treated as road segments without length. Depots (route origins) and destinations exist in a fine-mesh grid, independent from the road network graph, and each grid cell is treated as one charging infrastructure site in the cost model. The exact coordinates of rest stop parking areas are used to link these to all road segments within one kilometer (0.62 miles) distance. The term charging station may be used interchangeably with charging infrastructure placed at rest stops.

**Charging infrastructure (static, dynamic, offered, built).** Divided into infrastructure for charging vehicles while stationary (static charging infrastructure) and in motion (dynamic charging infrastructure). Includes costs for hardware, installation, maintenance, establishing a connection to the electrical grid, operational grid charges, and capital interest. All costs are summed over the infrastructure lifetime (discounting period) and annualized over the parameterized infrastructure lifetime. User prices are determined assuming the within-period

utilization will be sustained throughout the lifetime. This can lead to underpriced utilization early in the lifetime, if other competing charging infrastructure is built later, which makes the economic lifetime of the site much shorter than its expected discounting period. Such inaccuracies may affect charging behavior in the simulation but have less impact on estimations of total system cost. The model distinguishes between offered infrastructure and built infrastructure, where an offer means that the scenario definition permits infrastructure to be built in a location if it is in demand, and built infrastructure is the simulated outcome of a period that contributes to system cost.

**ERS network, electrification rate.** The electric road network is the subset of the entire road network where dynamic charging via ERS is supported. A scenario parameter for the ERS network is its electrification rate. An electrification rate of 40% means that physical ERS infrastructure is built on 40% of the total lane distance of the ERS network, interleaved by non-electrified gaps. If the power per vehicle is doubled and the electrification rate is halved, the effective rate of energy transfer per vehicle remains unchanged. ERS network sizes refer to bidirectional road distance, including gaps. Reducing the electrification rate reduces the cost of the ERS infrastructure but increases costs in other parts of the system.

**Route.** An unbroken one-way sequence of road segments describing a directed movement along the road network, from origin to destination. Each route is associated with metadata that describes the annual number of tons of goods transported and the number of truck transports, per vehicle weight class. Effective load factors vary, and as far as we can tell from the data provider, the number of transports includes trips without cargo. The first road segment in a route is always a depot, and the last is always a destination. The first time a route passes near a rest stop, the zero-distance rest stop is inserted into the route (but the truck may not stop there).

**Vehicles, trucks.** In the present study, MOSTACHI has been applied to simulate freight traffic with medium and heavy-duty trucks. However, the traffic data used contains no explicit information about individual trucks. Trucks only exist in the model as virtual agents that start at a depot, traverse a route (possibly including rest stops), and stop at a destination. Vehicles are always assumed to be equipped with the battery and vehicle technology of the current 5-year period for which the simulation is calculated: a departure from reality made to simplify the calculations. Vehicles that utilize ERS must be equipped with a separate ERS pickup, which adds cost.

**MGV16, MGV24, HGV40, HGV60.** Trucks in four weight classes are included, using definitions inherited from the input transport data. MGV16 is a medium-heavy two-axle goods vehicle (truck without trailer), total weight of 3.5–16 tons, which is usually used in local distribution. MGV24 is a medium-heavy three-axle truck without a trailer, total weight of 16–24 tons, which is usually used for construction work. HGV40 is a heavy two-axle tractor and three-axle trailer, total weight of 25–40 tons, which is usually used for long-distance transport. HGV60 is a heavy three-axle truck with a four–axle trailer, total weight 40–60 tons, which is used, for example, for transporting round timber[34].

**Battery pack.** Battery electric vehicles are equipped with battery packs, which are modeled separately from the rest of the vehicle, both parameter- and simulation-wise. The usable (net) storage capacity (in kWh) of vehicle battery packs is determined per route, vehicle class and time period, based on (a) the maximum output power per battery capacity for that simulation period, (b) the minimum required output power for that vehicle weight class, and (c) the capacity that minimizes the total cost of transporting the annual cargo. Conversions between usable (net) and installed (gross) storage capacity are made via the SoC

window parameter, which represents the fact that current battery chemistries perform so poorly near minimum and maximum SoC that these are artificially hidden by the manufacturer from vehicle users. Technical sources of values for battery parameters, such as weight per capacity, usually state gross figures.

**State of charge (SoC).** The energy level in a battery pack is referred to as the SoC, reported in percentage of net capacity or kWh. The SoC is updated sequentially for each road segment along a route. During dynamic charging, the SoC is updated based on the difference between supplied power and propulsion power. The SoC is initiated at 50% at the depot, followed by an opportunity for depot charging limited by the route's share of total daily operating time. Depot charging may not be available due to a lack of infrastructure or an incompatible charging strategy. If the energy level anywhere along the route reaches a minimum threshold for remaining range, the current combination of battery capacity and charging strategy is rejected.

**Charging strategy.** Along a route, charging infrastructure can be offered at many locations, resulting in different direct and indirect costs. Successful traversal of a route may require the utilization of all or a subset of offered charging locations, with outcomes dependent on battery capacity. The charging strategy defines which locations are considered: *NA_Diesel*: The vehicle is equipped with a diesel powertrain —charging infrastructure is ignored. *Depot*: The vehicle only charges at the depot. *DepotAndErsCharging*: The vehicle charges at the depot and ERS. Becomes ErsCharging if depot charging is not available on the route. *ErsCharging*: The vehicle charges only from the ERS infrastructure. *AllPlannedStops*: The vehicle charges at the depot, destination, and when the driver takes a legally mandated break after at most 4.5 h of driving. Breaks are taken up to 60 min earlier if charging is possible at a rest stop. Becomes PublicStaticCharging if depot charging is not available on the route. *PublicStaticCharging*: The vehicle charges only at rest stops and destinations. This strategy is not tested separately. *AllPlannedStopsAndErs*: As AllPlannedStops, but the vehicle can also charge from ERS.

A vehicle cannot charge at a type of charging infrastructure if no such infrastructure is offered along the route, regardless of the charging strategy. With all strategies, driver breaks are taken up to 30 min earlier if a rest stop is available, else a roadside stop takes place. This logic represents that the majority of real stops for driver rest do take place at designated rest areas (conclusion from private communication with Patrick Plötz, who had access to stop data from the major European truck OEMs for use in a study mapping and characterizing stop locations[35]).

**Charging power and C-rate.** When a vehicle is charging while stationary, this takes place at the lowest of the infrastructure's maximum power per vehicle, the battery's maximum allowed charging power, and the lowest power required to reach full charge before departure. When a vehicle charges dynamically from ERS, this rate is capped by the per-vehicle maximum power of the infrastructure minus power for propulsion, and further capped by what the battery can receive. The maximum charging rate of a battery is a parameterized (period-dependent) multiple of the effective C-rate, with C-rate being (input or output) power divided by storage capacity. This means only vehicles equipped with large battery packs may be able to fully utilize high-powered charging infrastructure, when such is available.

### Model calculations
This section describes how different cost components are calculated and how they are related.

**System cost.** The sum of all levelized costs of vehicles, batteries, charging infrastructure, capital interest, fuel/electricity, driver salaries,

and GHG emissions (internalized and externalized, or taxed and untaxed). Excludes road and pollution taxes. All costs are in 2020 Euros; no conversions are made to net present value or to adjust for inflation. Interest rates are considered rates above inflation.

**Transport cost.** The sum of all levelized costs affecting the vehicle operator—vehicles, batteries, capital interest, fuel/electricity, distribution, driver salaries, and all taxes. Residual value is subtracted for vehicles and batteries.

**Route traversal cost.** System and transport costs are calculated by simulating the traversal of each transport route independently. Traversal costs are then multiplied by the number of annual traversals and summed to calculate the annual total. Changes in powertrain can result in changes in the time taken to traverse a route or the cargo carrying capacity of trucks operating on the route. This will implicitly affect how many trucks are assumed to operate in the system, though, as real vehicles are assumed to operate on many routes, the simulation does not keep track of the individual vehicles making up the population.

**Weight adjustments.** Electric vehicles will differ in weight compared to the original diesel vehicles. Cargo carrying capacity adjustments are made to the per-route traversal costs to compensate for that a different number of annual trips are required to transport the same total amount of cargo with a maintained fill-rate. Loading capacity is often limited by cargo volume rather than weight, and vehicles often travel with less than full load—therefore, cargo capacity is modeled to change by 30% of the weight difference between ICET and BET. The remaining 70% of the weight difference affects energy consumption, with 10% weight change resulting in 5% change in energy consumption[36].

**Vehicle cost.** The sum of the purchasing price and maintenance costs minus residual value, levelized over the assumed economic lifetime. Vehicle cost excludes battery costs.

**Battery cost.** The purchasing price minus residual value, levelized over the simulated lifetime. The purchase price depends on the installed usable battery capacity and the net battery price in the current simulation period. Net battery price is the gross price (what is normally listed in sources) divided by the SoC window size.

Residual value depends on battery state of health (SoH) and year of decommissioning. Batteries degrade from passed calendar time and use, beginning at 100% SoH and ending at 0% SoH when the battery can no longer store any meaningful amount of energy. The market value of used BET traction batteries, at a reference SoH (80% in this study), has been approximated using an S-curve, from zero value in 2020 to 30% of the price of new batteries in 2050. This emulates a maturing second-life market and recycling industry. Batteries remain in vehicles until any of four conditions are met—the vehicle reaches its end of life, 80% SoH has been reached, remaining battery capacity is insufficient to complete the route, or the maximum power output is insufficient to meet the requirements of the truck. The most common threshold to be reached in later simulation years is the end of truck lifetime. Decommissioning before 80% SoH reduces the value loss proportionally to the improved health.

Battery ageing is approximated using the sum of calendar ageing —at a constant rate per calendar time—and cycle ageing. Cycle ageing takes place when energy flows into and out of the battery, calculated for each traversed road segment along a route as $a_{cyc} = \frac{1}{2} \cdot e \cdot \max[\frac{3}{4}, (c/c_{ref})^2]$, where $a_{cyc}$ is the percentage of capacity loss from cycle ageing, $e$ is energy flow (in kWh), $c$ is c-rate, and $c_{ref}$ is reference c-rate. C-rate is power divided by battery capacity, such that a c-rate of one means the battery is charged or discharged in 1 h. On a single simulated road segment (which includes stops), either charging or discharging takes place—batteries are conceptually like water tanks, and energy

can only flow in one direction at a time. Charging and discharging are capped at multiples of $c_{ref}$. The constant $1/2$ is included because a full cycle consists of both charging and discharging, and $3/4$ is a lower bound for the rate of cycle ageing. When total ageing sums to one, the battery has reached the reference SoH and is decommissioned.

This simple equation has no direct source and approximates what the main author has understood about battery ageing mechanisms from battery experts and literature (see references in the Supplementary Parameter Tables). The equation has been designed to (a) use variables available in the model, (b) allow parameterization of the aging rate, (c) let higher charging power result in more rapid battery ageing, and (d) let very low charging power still result in some ageing. No attempt has been made to capture ageing effects that are not primarily controlled by changes in charging infrastructure design, such as battery temperature. Capturing the influence of current battery SoC on battery ageing would likely improve results, but would necessitate much more complex charging strategies.

The battery aging model is a potential weak point in the study, as the modeling of battery ageing is still an active area of research. However, since most batteries are decommissioned at the end of the vehicle's lifetime, flaws of the battery ageing model should only impact the main conclusions if battery ageing is grossly underestimated.

**Charging infrastructure cost.** A parameterized linear model is used to calculate infrastructure cost, with a fixed cost per site and an increasing cost per kW of installed peak charging capacity (site total). Maintenance costs are assumed to be proportional to installation costs, applied yearly, and profit margins are applied. The initial cost of connecting to the power grid is included, as are operational grid fees. All costs are levelized over the discounting period for that type of infrastructure.

ERS installation costs are penalized by a risk multiplier for early simulation periods, to penalize construction before Europe has agreed on a common technical standard. Early construction could result in a need to later reinvest in infrastructure that has already been built. The risk is high in model year 2020 and then decreases rapidly.

**Charging infrastructure pricing.** The annualized levelized infrastructure cost of the charging infrastructure is spread across the annual sold energy within the simulated time period. Static charging fees are calculated per site, and ERS use is subject to a shared fee across the entire ERS network.

**Fuel or electricity cost.** Fuel costs (parameterized) are pre-calculated per simulation period based on assumed costs of fossil and renewable diesel fuel and the blend ratio between the two. AdBlue costs have not been included. The cost of electricity (parameterized) is specified as a night- and daytime tariff per electrical grid region. Separate utilization curves are used for each charging infrastructure type to calculate the mean electricity price for the entire day. This means that depot charging is closer to night-time prices (assumed lower), and the remaining charging types are closer to daytime prices (assumed higher).

**$CO_2$ emissions cost.** GHG emissions in the model originate from diesel consumption, electricity consumption, and battery production. Diesel emissions depend on the blend ratio of fossil and renewable diesel. Part of the social cost of carbon is internalized through a common carbon tax applied to emissions from fuel, electricity, and battery production. Taxing embodied fossil GHG emissions from battery pack production in this way raises battery costs by 5–10% with the parameter values used in this study.

**Post-hoc adjustment of renewable fuel ratio and $CO_2$.** The mixing ratio of renewable and fossil diesel is estimated in advance for each simulation period, but recalculated at the end of the simulation based

on an assumption that renewable fuels are maximally used up to a supply cap. This results in adjustments to the total costs of fuels and emissions and estimated emissions.

**Driver cost.** Applied per time unit, excluding time at the depot. We assume one driver per truck.

**Capital interest cost.** Interest is applied to all capital investments, assuming a flat amortization over the discounting period. Three different rates are applied depending on assumed financial risk, with the lowest rate used for state-backed ERS investments and the highest rate for depot charging infrastructure that is built for one company and cannot easily be moved.

**Taxes.** Today's taxes on diesel fuel have been separated to the best of our ability into $CO_2$ tax, pollution tax, and road tax, where the road tax is applied equally to diesel and electricity based on the HGV40 energy consumption rate for each simulation period. Road taxes are thus independent of the choice of powertrain. A separate pollution tax is applied in later simulation periods, on the assumption that society will want to penalize the additional particle and noise emissions of ICETs once BETs have become ubiquitous.

## Model and data validation
We have strived to ensure that MOSTACHI not only captures the desired system dynamics but also performs its calculations correctly.

Numeric variables with strongly typed units (e.g., kWh per kilometer) are used in place of generic floating point variables throughout the program source code to minimize the risk of erroneous calculations and variable assignments. Informal validation of the model has been performed by ensuring that input data passes through without distortion, that output values are reasonable in relation to each other, and that adjustments in one parameter have expected effects on other parameters. Input transport data has been validated in a separate report against national statistics, both at the population level and against measured traffic on the road network[22]. Parameter assumptions have been sourced from a mix of academic sources, gray literature, and industry experts. Experts have occasionally been asked to provide entire sets of parameters when these are known to be subject to trade-offs (e.g., selection of battery chemistry). Visualization techniques have been used extensively throughout the development to identify geographical and temporal anomalies in the input and output data.

It has been difficult to make informative comparisons with findings by other research groups, mainly because a core contribution of this work is to model system interaction effects as emergent properties rather than as fixed input assumptions. Such emergent properties include the share of traffic in different vehicle segments that have incentives to use ERS, the rate at which transport is electrified, the costs of charging via different types of infrastructure, and the selection of roads on which ERS should be built. In general, the results from simulation with MOSTACHI are more extreme than assumptions made in other research, representing winner-takes-it-all mechanisms that result from previously identified economies of scale and scope. Findings about battery capacity reductions resulting from ERS availability are strongly supported by prior research, and the mechanisms are further explained in this work.

Both vehicle purchase costs (Supplementary Fig. 1) and itemized and total levelized system costs (Supplementary Figs. 3 and 4) have been confirmed to be near the values presented in dedicated total cost of ownership studies comparing ICETs and BETs.

## Input global parameter values
A guiding principle for the development of MOSTACHI has been to avoid built-in assumptions about what a future electrified transport

system should look like, in particular, how electric vehicles of different classes and in different operational patterns should be charged. The model, therefore, uses a large number of numeric input parameters with separate values for each 5-year period. These represent expected battery technology advancements, fuel and electricity prices and emissions, charging infrastructure construction costs, vehicle costs and performance, and other properties that are independent of the system dynamics and for which there are generally much more reliable forecasts available than for what we aim to simulate.

Input parameters can be changed as part of defining scenarios to be simulated, which facilitates sensitivity and robustness analysis and exploration of how the system dynamics are influenced by different possible societal trends, such as increasing or decreasing density of traffic or relative pricing of electricity and diesel. Some parameters may require new values if the model is applied to other geographic areas or vehicle segments.

Parameter tables are provided as Supplementary Material and enclosed with the MOSTACHI source code.

### Input traffic patterns

Here follows a summary of the methodology used to prepare traffic pattern data for this study. Except for a few methodological improvements, detailed here, the full methodology, all validation metrics, and references to validation datasets are described in a prior technical report[22].

In the absence of directly measured traffic data, transport route data in the form of an origin-destination matrix was exported using the 2017 base scenario from the Samgods transport simulation[23]. Samgods is an independently developed and extensively validated Swedish national model of freight transport within, into, out of, and through Sweden. Input to the Samgods model is survey data capturing the annual flows of goods using all modes of transport (road, rail, sea, and air) between all pairs of Sweden's 290 municipalities, plus major foreign regions and key transport hubs such as ports. Samgods then distributes these goods flows on modes of transport, including rail, sea, and five different truck weight classes, of which we have retained four (MGV16, MGV24, HGV40, and HGV60) due to perceived data quality issues with the last (LGV3). The input dataset encompasses around 200,000 freight flows, further split into yearly tonnage per vehicle weight class. We observe that Samgods generally assigns goods to lighter vehicles on short routes and heavier vehicles on long routes, but refer to its documentation for details.

This base scenario dataset was shared with us by Magnus Johansson at the Swedish National Road and Transport Research Institute (VTI). Data in a similar format, capturing transport patterns across all of Europe, have since been made available by others[15], which should make it possible to apply this methodology to directly study effects across all of Europe.

To convert region pairs to routes on the road network, a point coordinate was sampled within each origin or destination region (hubs excluded), independently for each flow. Points were sampled based on openly available polygon data[37] representing commercial non-retail zoning areas in Sweden, under the assumption that most truck freight begins or ends within such an area. Polygons were weighted based on provided metadata about the type of industry, the number of businesses, and the number of employees within the areas. A note here is that during validation, a high-resolution raster of population density was used in place of the polygon data to sample origin and destination points. The switch to use commercial zones was motivated by that population density gives sampled points within residential rather than industrial areas within cities, which does not match how heavy-duty trucks are used. This switch should only affect within-city route selection and should have no impact on the results presented in this study.

After enriching the transport flows with start and end coordinates, the Open Source Routing Machine (https://project-osrm.org/) was utilized to determine the shortest route along the OpenStreetMap road network for each coordinate pair. This produced long sequences of very short segments, together representing unbroken routes. The resulting graph, made up of all these segments, was simplified by merging sets of segments that formed a non-branching chain into one segment, retaining (and combining) metadata about segment length, speed limits, directionality, and lane count. Potentially useful road network properties such as curvature, road gradient, and number of lanes were not available with sufficient quality in OpenStreetMap to be of use.

As the input dataset only contains traffic that involves the Swedish road network, it has not been possible to calculate any charging infrastructure economics outside of Sweden. Routes have been truncated at the border in the simulation, and the route distance outside the border has been assumed to have access to charging infrastructure with the same density, ratios, and pricing as the part inside Sweden.

Validation and calibration of the resulting route dataset were performed against (a) national statistics for total transport assignments, total moved tonnage, and total driven distance, (b) statistics on truck fill rates, and (c) interpolated measurements of real traffic throughout the road network. There are minor but important differences in what traffic and vehicles are included in the Samgods data and in the national reference datasets, which complicates the validation work. Validation against road traffic took place by aggregating the route data into annual vehicle passages per road segment, then by further computing the maximum traffic intensity of any segment intersecting each cell in a fine-mesh spatial grid. The same rasterization was performed for official interpolated measurement data for the entire Swedish road network, and the two rasters were visualized as geographic bar charts to study spatial correlation and identify geographic discrepancies.

Two calibration steps were taken: (1) to increase the number of trips made per route to more closely match the real distribution of load factors (which effectively introduces return trips without load), and (2) to jointly rescale number of trips and tonnage per route by common scale factors for all routes in different length buckets, such that the sum of traffic in all buckets more closely matched real measured traffic on the road network. After calibration, the distribution and volume of traffic along the road network closely matched real numbers, except for underestimates of total traffic within major urban centers. We speculate that this is explained either by bus traffic—absent in Samgods but present in measured traffic—or by that Samgods undersamples short-distance transport assignments. We have not come up with any way to confirm or reject these hypotheses. Total travel distance and number of trips were then 40% above national statistics, and total tonnage 10% above national statistics, which we believe is approximately correct given that the statistics—but not Samgods and measured traffic—exclude foreign-registered trucks, known to make up a substantial ratio of total traffic but a smaller ratio of total cargo tonnage.

The transport pattern input data lacks information about the temporal distribution of when routes are traversed, and which routes are traversed by the same vehicles and in what order. It is partly because of this limitation that cost optimization takes place per route. Furthermore, this limitation leads to charging infrastructure utilization rates (ratio of peak installed power utilized per hour of the year) being estimated globally per charging infrastructure type and given as an input to the simulation. The same holds true for vehicles—as no information is available to the model about individual vehicles across transport routes, economic calculations in this study cannot account for effects due to variations in vehicle utilization within the vehicle population, from requirements to traverse routes with very differing properties, or operation during unusual times of the day.

While low-bias GPS (global positioning system) data from well-sampled large vehicle fleets could potentially yield a more accurate

model of today's transport system than the route data employed here, historical data would not account for changes in transport planning (assignment of goods to vehicles) arising from changed conditions along the road network. Furthermore, GPS data is difficult to access for researchers and transport planners worldwide, and methods requiring this data may be more challenging to scale. Furthermore, if unbroken GPS traces are accessible, it is straightforward to transform such data into the route format used in MOSTACHI, and the model could with relative ease be adapted to handle multi-stop routes. In short, route-based analysis fails to capture some of the operational characteristics of commercial trucks and truck fleets, but historical GPS data would only partially resolve this.

### Input order of ERS deployment

The utility of ERS infrastructure depends both on where on the road network the infrastructure is placed and the size of the ERS network. Maximizing the lifetime utility of an ERS network requires not only that an optimal layout of the network is determined, but also that the step-by-step order of installation is optimal. We do not attempt to solve this very difficult optimization problem, but we compute a build-out order that achieves greater utilization rates than the naïve solution of selecting the roads with the greatest AADT. When a MOSTACHI scenario is defined to include an ERS network of size X km, this pre-computed buildout order is used to select on which segments of the road network ERS should be offered. A subset of offered road segments is then selected for ERS construction during the demand-driven placement of infrastructure.

The expansion order used for ERS in the present study has been calculated using Supplementary Method 1. This order maximizes synergies between already built ERS and new ERS, but does not take static charging infrastructure into account. The definition of synergies here is that the next road segment to build is that which is most traversed by vehicles using the already built segments. The starting point of the ERS network was set in the south of Sweden, to maximize the synergy effects with electric roads in neighboring countries. See Supplementary Fig. 5 for a map of the ERS build-out order.

### Input rest stop locations

Independent prior analysis based on GPS data from connected trucks[35] has identified 424 locations within Sweden where many trucks from several manufacturers stop. These locations, primarily rest areas along major motorways, are used as options for the placement of public fast charging stations. A map of the locations is available in a prior technical report[16].

No pre-computation of build-out order was performed for static charging infrastructure, which is instead offered in random order (the same order for all experiments). While it is likely that sort orders achieving greater utility can be found, centralized coordination of privately financed investments in charging infrastructure was not deemed a realistic approximation of reality.

### Output data

Simulation of a MOSTACHI scenario results in the generation of three data files containing columnar tab-separated values.

Data file one contains a summary of key metrics for the entire simulated system, including total $CO_2$ emissions, total annual system cost, total annual cost components for ICET and BET traffic, share of electrified transport work per vehicle class, total energy delivered per type of charging infrastructure and charging strategy, and total cost per type of charging infrastructure.

Data file two contains information about the location, type, installed capacity, utilized capacity, and cost of use of each charging infrastructure object present in the simulation. Data for the ERS infrastructure is reported in a $1\,km^2$ raster, while the static charging infrastructure has an exact coordinate per site.

Data file three contains total annual traffic (vehicle passages and tons) per cell in a $5\,km^2$ raster, disaggregated by vehicle class, driveline, battery capacity, and charging strategy.

Although the simulation runs in 5-year periods, output values represent annual data. MOSTACHI only generates simulation results. It is up to the user of the tool to run further analysis of these datasets to answer research questions of interest.

### Reporting summary

Further information on research design is available in the Nature Portfolio Reporting Summary linked to this article.

## Data availability

The input datasets and raw simulation output logs generated in this study have been deposited in the Zenodo database under accession code 15755307[38]. The reorganized data used to draw the figures in this study are provided in the Source Data file. Source data are provided with this paper.

## Code availability

The source code for the version of MOSTACHI used in this study has been deposited in the Zenodo database under accession code 15755307[38]. The latest version of MOSTACHI is available on GitHub [https://github.com/JakobRogstadius/MOSTACHI/].

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

## Acknowledgements

This research was funded by Region Blekinge and the European Regional Development Fund through the project "Genomförbarhetsstudie av elvägspilot E22."

## Author contributions

J.R. contributed to ideation, development of MOSTACHI, design of experiments, data preparation, data analysis, and writing of the manuscript. H.A. contributed to cleaning, preparation, and validation of transport flow data. H.S. contributed to the design of experiments and to the writing of the "Introduction," "Results," and "Discussion." K.F. contributed to preparing MOSTACHI for public release.

## Funding

## Competing interests

The authors declare no competing interests.
