## [Transparent Peer Review file · Nature Communications]

Correcting Market Failure for No-Regret Electric Road Investments under Uncertainty

Corresponding Author: Dr Jakob Rogstadius

Version 0:

Reviewer comments:

Reviewer #1

(Remarks to the Author)

This paper evaluates the potential pathways for decarbonising heavy goods vehicles using a newly developed model for studying the impacts and interactions in the road freight industry.

The research is good and provides a solid examination of the system costs and ghg emissions, and provides interesting and valuable insight into the trade offs between charging and battery sizes.

My main criticisms of the paper is with the structure and presentation, and it needs a restructure to make it easier to follow.

I think it's reasonable to put the MOSTACHI outline towards the end of the paper, but more explanation should be made in the main paper to clearly outline the scenarios and then explain the results in the figures.

Further explanation in the model validation should be made to highlight clearly what the validation was made against - some references to the compared research would be helpful.

The figure captions contain too much text. The long descriptions should be incorporated into the body of the paper and some spacing into the presentation of results.

Figure 2 would be better split into 3 separate figures ((i) CO₂ & cost, (ii) energy demand, (iii) charging demand). This would enable the charts to be larger and easier to read/inspect.

On page 5, some justification should be added on the selection of charging power and battery sizes (particularly for the different vehicle types).

Reviewer #2

(Remarks to the Author)

This paper reports on a serious and complex study of the benefits of investing in electric road infrastructure for electric trucks. Interesting results are presented, notably on greenhouse gas emissions, battery demand (and by extension the environmental effects of electrifying trucks, which go beyond carbon) and the impact of different scenarios more or less favorable to road electrification on infrastructure use, making it possible to extract policy recommendations to encourage the electrification of mobility.

The paper is of good quality, so my report will be short. In my opinion, some points need a little clarification, mainly in the description of scenarios and their analysis, but I have no doubt that this is more a question of a desire for synthesis than of scientific shortcomings. Once these points have been clarified, I would recommend the publication of this interesting paper.

Here are the main points that I feel need to be clarified:

General comment: There is no clear description of how the evolution of oil prices and, more broadly, carbon prices are

considered, even though road electrification is a project that will take several decades to complete, so these prices are bound to change over such a long period.

Line 120: « with 40% coverage », why this value of 40% ?

Line 123-130: About charging infrastructure at depot/rest stop/destination, is it assumed that the infrastructure is sufficient to always have an available charging station if the truck wants to charge? If yes, it's a strong assumption, it must be stated.

Line 131: 375 kW for an electric road seems to me to be a very optimistic hypothesis (but it may be a lack of knowledge on my part, in which case my apologies). If it is as optimistic as I think, it would be useful to point it out, and to specify how this hypothesis impacts your results: are they still valid with lower charging powers?

Line 132-134: Today, a li-ion battery has an energy density of around 200Wh/kg. So, between a 75kWh battery and a 1000kWh battery, there's a difference of almost 5 tons for the vehicle to carry, which has an impact on fuel consumption. Has this increase in consumption been considered?

Line 161-163: "As scenarios with ERS include more infrastructure than scenarios without, ERS accelerates both electrification and the resulting reductions in mean cost and GHG emissions.". If I understand correctly, you are assuming in your model that investments in ERS are in addition to static infrastructure, and not that there is a choice to be made between investments in ERS and investments in charging stations. Or are you assuming that with ERS the necessary investment in batteries is lower, so more can be spent on the charging infrastructure, and at the end the overall cost is lower? This point should be clarified.

Reviewer #3

(Remarks to the Author)

The paper is a great step in explaining the included simulation tool and showcasing the benefits of the ERS at the same time. Given the large amount of content in the paper, it is suggested at this stage to first organise it better for more clarity.

1. The methodology in the abstract could be clearer - perhaps one line summing up what was done in the paper would help.
2. The order of the paper seems unusual - readers are used to reading research papers in a certain order and it is good to write in that order. Hence, perhaps the methodology could be explained first followed by the results. For example, the content could start by explaining the need for a simulation tool, what is MOSTACHI, what are the inputs and outputs (how to use), how it works (methods section), and then an applications or examples section with each of the results figures and the methodology behind how they were generated. The paper could then be ended with observations from these results and a discussion on the ERS.
3. Captions for figures are too long in general. Most of it could go into the text, especially the results figures (2 to 7). All this content along with the methods on how these figures were generated could go in a separate section after the methodology of MOSTACHI is explained. Fig 2 has too much content resulting in small text sizes.- it could be split into multiple figures.
4. Explaining the model of MOSTACHI needs a proper flowchart diagram. Fig. 1 attempts to do this but is unclear because of a collection of images on the left, a very simplified flowchart on the right and a large amount of text in the caption. This could be improved and a more cohesive self-explanatory flowchart in its place would make it much better.
5. The transition phase is an important one to be considered and should be a key part of the tool. When it comes to implementing the ERS, questions such as emissions reduction, cost breakeven and reduced battery sizes in vehicles have been answered already. The two major barriers now are deciding who takes up the responsibility, and having enough incentive for the one who takes it. At this stage, it seems likely that fleet operators (who would benefit most from ERS) would need to push the government to take it up, and the government should have the incentive (say, in terms of tax revenue considering that fuel tax would be lost) to support the ERS. Major pilots for the ERS are now stuck at the trial stage because of lack of government support in getting the right investments. It would be interesting to see how these barriers may be solved using the tool.
6. Consequently, the tool may also consider more perspectives such as that of the infrastructure provider and government, as infrastructure rollout should keep their pockets happy too (and policy would have a huge impact on the rollout).
7. The major input for the tool is traffic data. It is unclear if it just needs traffic flows/counts or journey origin-destination pairs. While Sweden has Samgods, most countries only have traffic counts (AADTs or flows) and it is extremely difficult to predict origin-destination pairs based on just this data. Therefore, if MOSTACHI only needs the traffic counts, it is easier. However, then the question becomes about how the tool uses this data to predict origin-destination pairs.
8. It may be worth considering splitting the paper into a report or manual about the tool and a research paper answering the research questions separately (including point 5).

Thank you for a great job in initiating the development of this useful tool. I hope that the paper would benefit from reorganising and these few fundamental questions being solved may help make the tool more accurate.

Version 1:

Reviewer comments:

Reviewer #1

(Remarks to the Author)

Thank you for making these changes, I am happy with the revised structure.

It should be accepted for publication.

(Remarks on code availability)

Reviewer #3

(Remarks to the Author)

Most of the comments from the previous round have been answered well. Some additional comments are noted below.

1. Hybrid or range extended electric vehicles, where biofuel or hydrogen driven engines may be used in operations where charging infrastructure is not feasible, may be an alternate option that could be considered and pushed for by vehicle operators. The costs of such types of vehicles will also affect charging infrastructure usage. It is suggested to either consider this or add it as a limitation of the model.

2. Change in cargo carrying capacity of an electric vehicle vs diesel may not just result in an increase in the annual trip count but could also instead result in an increase in the number of vehicles. This has an effect on the vehicle capital cost which increases along with driver cost.

Additionally, it depends on the commodity being carried, as the vehicle has two kinds of limits – axle load limits and gross vehicle weight limit. Therefore, it may be possible for some types of freight to be loaded non-uniformly in the trailer to meet the axle load limits and reach the gross vehicle weight allowance. This causes a different change in carrying capacity as loads that can be carried currently in diesel 6x4 trucks may not be possible in electric 4x2s even if they are not currently mass-limited.

3. Papers such as <https://doi.org/10.1109/ITSC57777.2023.10422258> suggest that it may be beneficial to have split rest stops compared to one continuous rest stop as is also permitted in EU regulations. Changes in rest stop regulations will affect battery sizes and charger usage and should be addressed by the model.

4. Along with improvements in battery technology, improvements in road and tyre technology may result in higher axle load limit allowances, which would mean smaller payload losses and larger batteries possible in trucks.

5.1. The energy consumption values (kWh/km) for all categories of trucks do not seem to be dependent on the payload as seen in the supplementary information table. These values would be different for an empty and loaded truck (in the same category) and would be dependent on the vehicle's mass.

5.2. Furthermore, it is assumed that they reduce over time (is it by 1% every year)? This assumption might make sense for ICEVs as powertrain efficiency keeps increasing (as they are not very efficient currently ~40%), but BEV powertrains are already around 90% efficient... hence, similar efficiency improvements may not be expected. What may improve is the energy density of batteries, resulting in smaller battery sizes and reduced weights and theoretically reduced energy consumption, but it wouldn't in reality as operators would then carry more payload up to the weight limit to reduce logistics cost.

5.3. Additionally, the energy consumption depends highly on the route taken because of factors like road gradients and traffic.

All these assumptions should be reconsidered as the study is very sensitive to energy consumption.

6. Rather than directly having the Results section after the Introduction, it is suggested to split it as Introduction, Research Questions, Methodology (not the full MOSTACHI explanation, but just an explanation of how MOSTACHI was used or set up – perhaps the Experiment Design bit) and then the Results/Discussion, followed finally by the current 'Methods' section which could also be renamed to 'Details of the Simulation Tool (MOSTACHI)' or something similar. This would probably make it easier to go through as seeing the Results after the Introduction and Methods at the end is undeniably hard to figure out.

(Remarks on code availability)

The readme file is quite comprehensive and it seems straightforward to set up the code.

Version 3:

Reviewer comments:

Reviewer #1

(Remarks to the Author)

The paper has been revised significantly, beyond the peer review, and should be accepted for publication

(Remarks on code availability)

Reviewer #3

(Remarks to the Author)

Thank you for addressing the major comments well. Some minor comments are included below, primarily regarding some of the responses, without needing any major changes in the methodology.

1. Regarding range-extended vehicles: It would be good to be able to simulate those journeys away from charging

infrastructure, particularly in countries with such journeys such as Australia. This could be added in a future update to the software so that it is capable of simulating more countries.

2. Regarding axle configuration: Currently the highest weight category (44T) vehicles are 6x2 vehicles. In BEVs, manufacturers place batteries between the front and rear axles of the tractor unit. This takes up some space in the wheelbase, which means that the length of a 4x2 BEV would be the same as a 6x2 ICE truck (16.5 metres). There is a maximum limit on the combination length (tractor+trailer), which is 16.5 metres. A 6x2 (or 6x4) BEV exceeds this limit due to the increase in the wheelbase of the tractor caused by the battery packs. Therefore, under current regulations, it is not possible to use a 6x2 or 6x4 articulated BEV on the road with a standard trailer even if manufacturers may have made such trucks. This is important to account for as it would need a major change in regulations to allow this to happen.

(Remarks on code availability)

RESPONSES TO REVIEWERS

Reviewer #1 (Remarks to the Author)

This paper evaluates the potential pathways for decarbonising heavy goods vehicles using a newly developed model for studying the impacts and interactions in the road freight industry. The research is good and provides a solid examination of the system costs and ghg emissions, and provides interesting and valuable insight into the trade offs between charging and battery sizes. My main criticisms of the paper is with the structure and presentation, and it needs a restructure to make it easier to follow.

Comment 1: I think it's reasonable to put the MOSTACHI outline towards the end of the paper, but more explanation should be made in the main paper to clearly outline the scenarios and then explain the results in the figures.

- The introduction has been shortened. Save for one paragraph, all information about MOSTACHI has been moved to Methods. Limitations have been moved to Discussion. The scenarios being analyzed are defined in the first section of Results and the scenarios have been better motivated.

Comment 2: Further explanation of the model validation should be made to highlight clearly what the validation was made against - some references to the compared research would be helpful.

- The model validation section of Methods has been improved.

Comment 3: The figure captions contain too much text. The long descriptions should be incorporated into the body of the paper and some spacing into the presentation of results.

- Figure captions have been shortened and the information has been incorporated into the main text.

Comment 4: Figure 2 would be better split into 3 separate figures ((i) CO₂ & cost, (ii) energy demand, (iii) charging demand). This would enable the charts to be larger and easier to read/inspect.

- The figure has been split into three parts.

Comment 5: On page 5, some justification should be added on the selection of charging power and battery sizes (particularly for the different vehicle types).

- The selections of charging power and battery sizes have been justified in Experimental Design. Higher charging powers and larger battery capacities have been added and experiments have been rerun. The primary effect of this is to increase the share of electrified traffic in early years without access to ERS.

Reviewer #2 (Remarks to the Author):

..., The paper is of good quality, so my report will be short. In my opinion, some points need a little clarification, mainly in the description of scenarios and their analysis, but I have no doubt that this is more a question of a desire for synthesis than of scientific shortcomings. Once these points have been clarified, I would recommend the publication of this interesting paper. Here are the main points that I feel need to be clarified:

Comments 1-5: General comment: There is no clear description of how the evolution of oil prices and, more broadly, carbon prices are considered, even though road electrification is a project that will take several decades to complete, so these prices are bound to change over such a long period.

- We actually found in other work (ref), to my own surprise, that electrification of most of the TEN-T Core network could most likely from a logistical point of view be completed in less than five years. Bureaucracy and slow planning processes seem to be the bottleneck. Inclusion of ERS in AFIR during the 2026 revision would likely be necessary for an EU ERS network to become reality.
- Altering the cost difference between electricity and diesel alters how much traffic is electrified, but not which charging infrastructure is preferred. It has therefore not been the focus of the paper, to maintain focus. Increasing the difference has minimal effect when electrification is held back by charging infrastructure access. The different scenarios show how different rates of electrification affect other metrics of interest.

Comment 2 Line 120: « with 40% coverage », why this value of 40% ?

- The selection of ERS coverage have been motivated in Experimental Design.

Comment 3: Line 123-130: About charging infrastructure at depot/rest stop/destination, is it assumed that the infrastructure is sufficient to always have an available charging station if the truck wants to charge? If yes, it's a strong assumption, it must be stated.

- It has been further clarified in Experimental Design how the availability of charging infrastructure varies between conditions.

Comment 4: Line 131: 375 kW for an electric road seems to me to be a very optimistic hypothesis (but it may be a lack of knowledge on my part, in which case my apologies). If it is as optimistic as I think, it would be useful to point it out, and to specify how this hypothesis impacts your results: are they still valid with lower charging powers?

- The selection of ERS power have been motivated in Experimental Design.
- The impact of ERS power is explored in figures 2 and 5.

Comment 5: Line 132-134: Today, a li-ion battery has an energy density of around 200Wh/kg. So, between a 75kWh battery and a 1000kWh battery, there's a difference of almost 5 tons for the vehicle to carry, which has an impact on fuel consumption. Has this increase in consumption been considered?

- The impact of total vehicle weight on vehicle energy consumption has been added to the simulation model, at a 5% change in energy consumption resulting from a 10% change in vehicle mass. As most of road freight is volume-limited rather than weight-limited, and as much of total distance is traversed with less than full load, 70% of the change in empty-vehicle weight is assumed to influence energy consumption, with 30% assume to influence cargo carrying capacity per route traversal. This is now explained under Methods > Model calculations > Weight adjustments.

Comment 6: Line 161-163: "As scenarios with ERS include more infrastructure than scenarios without, ERS accelerates both electrification and the resulting reductions in mean cost and GHG emissions." If I understand correctly, you are assuming in your model that investments in ERS are in addition to static infrastructure, and not that there is a choice to be made between investments in ERS and investments in charging stations. Or are you assuming that with ERS the necessary

investment in batteries is lower, so more can be spent on the charging infrastructure, and at the end the overall cost is lower? This point should be clarified.

- All explanations of the inner working of MOSTACHI have been moved together under Methods. We have tried to clarify the differences between offered and built charging infrastructure and how the composition and installed charging capacity of infrastructure is an emergent property in the system (i.e., “choices” are made as part of the optimization logic, but primarily in terms of installed charging capacity, not number of built sites). Battery capacity per vehicle is also an emergent property of the system that is determined through optimization. We believe the updated flowchart helps convey the simulation logic better.

Reviewer #3 (Remarks to the Author):

The paper is a great step in explaining the included simulation tool and showcasing the benefits of the ERS at the same time. Given the large amount of content in the paper, it is suggested at this stage to first organise it better for more clarity.

Comment: The methodology in the abstract could be clearer - perhaps one line summing up what was done in the paper would help.

- The method description in the abstract has been improved.

Comment 2: The order of the paper seems unusual - readers are used to reading research papers in a certain order and it is good to write in that order. Hence, perhaps the methodology could be explained first followed by the results. For example, the content could start by explaining the need for a simulation tool, what is MOSTACHI, what are the inputs and outputs (how to use), how it works (methods section), and then an applications or examples section with each of the results figures and the methodology behind how they were generated. The paper could then be ended with observations from these results and a discussion on the ERS.

- All content has been reorganized to match the journal’s guidelines.

Comment 3: Captions for figures are too long in general. Most of it could go into the text, especially the results figures (2 to 7). All this content along with the methods on how these figures were generated could go in a separate section after the methodology of MOSTACHI is explained. Fig 2 has too much content resulting in small text sizes.- it could be split into multiple figures.

- Figure captions have been shortened and the information has been incorporated into the main text.

Comment 4. Explaining the model of MOSTACHI needs a proper flowchart diagram. Fig. 1 attempts to do this but is unclear because of a collection of images on the left, a very simplified flowchart on the right and a large amount of text in the caption. This could be improved and a more cohesive self-explanatory flowchart in its place would make it much better.

- The old figure has been replaced with a new flowchart, which contains much more information about the computational steps taken within MOSTACHI.

Comment 5. The transition phase is an important one to be considered and should be a key part of the tool. When it comes to implementing the ERS, questions such as emissions reduction, cost breakeven and reduced battery sizes in vehicles have been answered already. The two major barriers now are deciding who takes up the responsibility, and having enough incentive for the one who takes it. At this stage, it seems likely that fleet operators (who would benefit most from ERS) would need to push the government to take it up, and the government should have the incentive (say, in terms of

tax revenue considering that fuel tax would be lost) to support the ERS. Major pilots for the ERS are now stuck at the trial stage because of lack of government support in getting the right investments. It would be interesting to see how these barriers may be solved using the tool.

- It is unclear what the reviewer is asking for, but we have tried to strengthen the text in Discussion that touches on policy implications.

6. Consequently, the tool may also consider more perspectives such as that of the infrastructure provider and government, as infrastructure rollout should keep their pockets happy too (and policy would have a huge impact on the rollout).

- User prices for all charging are set with expectations to recoup cost (at the time of construction), including profit margins and operational expenses. Competing infrastructure that is established later may jeopardize this cost recouperation, which is reflected in total system costs. Exploring how to ensure the governments' pockets are kept happy through policy intervention is the main contribution of this paper. We have hopefully conveyed this better by improving the structure and presentation of the content.

7. The major input for the tool is traffic data. It is unclear if it just needs traffic flows/counts or journey origin-destination pairs. While Sweden has Samgods, most countries only have traffic counts (AADTs or flows) and it is extremely difficult to predict origin-destination pairs based on just this data. Therefore, if MOSTACHI only needs the traffic counts, it is easier. However, then the question becomes about how the tool uses this data to predict origin-destination pairs.

- We have clarified in the introduction that data of the type required to run MOSTACHI is available for all of the EU.

8. It may be worth considering splitting the paper into a report or manual about the tool and a research paper answering the research questions separately (including point 5).

- All information about MOSTACHI has been collected in Methods.

Thank you for a great job in initiating the development of this useful tool. I hope that the paper would benefit from reorganising and these few fundamental questions being solved may help make the tool more accurate.

RESPONSE LETTER

REVISION 2

General changes in revision 2

In addition to changes in response to reviewer comments, the following changes were made to the manuscript in revision 2. All changes to the main document have been tracked and should show up with color highlights.

1. The abstract was improved to more clearly report the quantitative findings.
2. All mentions of BEV (battery electric vehicle) and ICEV (internal combustion engine vehicle) were changed to BET (battery electric truck) and ICET (internal combustion engine truck), for consistency.
3. Cumulative effects on reduced system cost, reduced GHG emissions and battery resource consumption are now reported quantitatively for figure 1 and figure 3, both in the figure captions and the accompanying main text.
4. The paragraphs about limitations of the study were extended and moved from Discussion to the beginning of Methods, to make the discussion easier to follow.
5. We removed the section discussing socio-economic return on investment (ROI) for different types of charging infrastructure at different stages of the transition. This removal is motivated by that the figure was deemed to be easy to misinterpret. The figure was also rather sensitive to changes in input parameters and we were unable to convey all the necessary nuances in the limited space. Instead, the ROI section was replaced with a new more thorough sensitivity analysis, using Monte Carlo methods. This section serves to remove any doubts that the main scenarios explored in the study were cherry-picked, even unintentionally.
6. Since last revision, the MOSTACHI simulation tool was applied for a different publication (<https://urn.kb.se/resolve?urn=urn%3Anbn%3Ase%3Ari%3Adiva-76076>). Learnings from that parallel work have been applied to this research. All simulations have thus been rerun and the figures and text updated to reflect any changes in simulation outcomes. The following substantial changes were made:
 - a. Two bugs were identified in the calculations of battery ageing and of operational costs for electrical grid connections. The source code implementation now matches the logic described in the documentation.
 - b. A few model parameters were fine-tuned with better sources. The only significant parameter change is that the social cost of carbon now follows the European Investment Bank's recommended shadow cost of carbon, which increases to higher levels than previously assumed. This has increased the cost of using fossil diesel in later years in the simulation.
 - c. Our understanding has improved of the underlying system dynamics that are being simulated. An attempt was therefore made to streamline the narrative in this manuscript, to focus only on dominant effects that have substantial impact, and to remove distracting remarks of less importance. Edits and clarifications to the prose have been made throughout the main text, but the structure and main message are unchanged.
 - d. The simulated implementation of "policy that encourages ERS use" was changed from "all BETs that drive on ERS must charge from ERS" to "the price of charging from ERS is capped at approximately the same level as the average levelized cost of public station fast charging at rest stops, and if this is not enough to fully cover costs in some years, ERS charging is subsidized". This change also followed the above-mentioned

- study, as this policy implementation was considered more realistic by contacts at the Swedish Transport Administration.
- e. Charging infrastructure utilization rates (an input parameter) were reduced for earlier simulation years, which has the effect of increasing the levelized cost of charging infrastructure, which in turn delays transport electrification. This logic is explained in a comment to Supplementary table 13.
 - f. A new “Logistic BEV penalty” parameter was added (Supplementary table 1), which acts as a multiplier for levelized transport cost with BETs. This simulates market resistance in early years to adopt BETs.
7. To summarize, we recommend a complete read-through of the Abstract, Introduction, Results and Discussion. Only minor changes were made to Methods, which are individually mentioned below.

RESPONSES TO REVIEWERS

EDITOR

Comment via email: *“The previous rebuttal letter looked too simple, which resulted in a doubted evaluation for your revision.”*

Response regarding poorly documented changes in revision 1: In the previous revision of this manuscript, comments had been left in the uploaded MS Word document to explain changes that had been made to different sections of the paper. The comments were meant to complement the individual responses we gave to the reviewers in revision 1, but we suspect the comments were stripped out by the submission handling system and thus were never passed on to be read by the reviewers. We apologize for this misunderstanding on our part, which we understand made it difficult to follow what changes had been made. We list these stripped-out comments from revision 1 below.

1. Reviewers had concerns about the general structure and clarity in the manuscript. Therefore, all report sections have undergone editing for clarity, with changes to the prose while keeping the meaning identical. Section titles and ordering have been changed to match Nature Communications’ formatting instructions. We were not able to use “track changes” functionality for this, as all text became marked as changed when paragraphs and sections were moved around.
2. The scenario “Triple-renewables” was replaced with “Capped-static”, as we believe the new scenario is more informative for readers. Text and figures referring to the replaced scenario have been updated accordingly.
3. Section *Transport Electrification, System Cost and Greenhouse Gas Emissions*: This section has largely been rewritten, after reflection and to better convey the main findings around costs, GHG emissions and share of electrified traffic.
4. Section *Discussion*: Limitations were moved to the discussion section.
5. Section *Methods - Summary*: The methods summary in Introduction was shortened and the flowchart of computation in MOSTACHI was moved to Method.
6. *Figure 8*: The flowchart was redrawn to convey more information about the model, its dependencies and its cost calculation logic.
7. *Methods – Model Calculations – Weight adjustments*: Changes in gross vehicle weight resulting from switching from ICEV to BEV now also impact vehicle energy consumption. Previously, weight only (partially) contributed to a change in the number of annual trips along a route required to move the amount of annual cargo.
8. *Methods – Model Calculations – CO₂ emissions cost*: Included embodied battery GHG in taxable emissions.
9. *Model and data validation*: This section has been expanded.

10. *Algorithm 1*: Pseudocode indentation was corrected.

REVIEWER 1 (REMARKS TO THE AUTHOR)

The reviewer requested no changes and recommended publication of the paper. We would like to thank the reviewer for their helpful and constructive suggestions in the previous round which helped to improve the manuscript.

REVIEWER 3 (REMARKS TO THE AUTHOR)

Comment 1: *“Hybrid or range extended electric vehicles, where biofuel or hydrogen driven engines may be used in operations where charging infrastructure is not feasible, may be an alternate option that could be considered and pushed for by vehicle operators. The costs of such types of vehicles will also affect charging infrastructure usage. It is suggested to either consider this or add it as a limitation of the model.”*

Response regarding hybrid trucks: The section *Methods – Model Capabilities and Limitations* has been updated to more clearly state that only diesel and battery electric trucks are simulated, not plug-in hybrids, fuel-cell electric, hydrogen combustion or methane combustion trucks. In this section, we have also mentioned hybrids as a possible direction for future work and commented on how we believe this addition would affect simulation results (very little).

Regardless, we argue that a future in which a significant number of hybrid diesel-electric trucks use ERS is unlikely. The European truck manufacturers have essentially ended all investment into combustion engine R&D (source: general comments about the industry from senior contacts at Scania CV AB over the past years – Scania will not offer hybrids after 2030). Second, ETS2 is going to substantially increase fossil fuel costs in Europe and we cannot see how a hybrid could be cheaper to operate than a BEV by the time ERS can be in place. Reference 7 in the main text also shows how it is very unlikely that any sustainable drop-in fuels will become both affordable and available in large quantities. Together, we consider these to be strong indications that hybrids will not be a longterm solution for road freight.

We also argue that incorporating hybrids in the experiments is unnecessary to answer our research questions. To the best of our understanding of the simulated system dynamics, ERS outcompetes static charging on levelized cost where it is built, while static charging competes with diesel outside the ERS network. This means that adding hybrids as a third alternative would either increase the utilization of ERS (a switch from diesel to hybrid) or reduce the utilization of public static charging (a switch from BET to hybrid). We cannot think of any reason why a hybrid truck in the simulated system would achieve lowest cost using only static charging and diesel, while if it was fully electric, it would also benefit from utilizing ERS. Therefore, we are quite confident that support for the main conclusions would become stronger, not weaker.

There is a possibility that hybrids could be motivated if a single truck needs to operate on multiple routes, some of which lack charging infrastructure. Unfortunately, we cannot simulate this due to limitations in the available input data. It could also be possible to reallocate these transport missions between trucks.

Comment 2: *“Change in cargo carrying capacity of an electric vehicle vs diesel may not just result in an increase in the annual trip count but could also instead result in an increase in the number of vehicles. This has an effect on the vehicle capital cost which increases along with driver cost.*

Additionally, it depends on the commodity being carried, as the vehicle has two kinds of limits – axle load limits and gross vehicle weight limit. Therefore, it may be possible for some types of freight to be loaded non-uniformly in the trailer to meet the axle load limits and reach the gross vehicle

weight allowance. This causes a different change in carrying capacity as loads that can be carried currently in diesel 6x4 trucks may not be possible in electric 4x2s even if they are not currently mass-limited.”

Response regarding changes in truck carrying capacity: The calculation requested by the reviewer is already incorporated in the simulation – changes in carrying capacity result both in changes in annual trip count and vehicle count. *Methods – Model Calculations – Route traversal cost* has been added to explain this more clearly. The handling is implicit, as trucks are assumed to operate on many routes and as our input data is per-route and not per-truck. Costs of traversing a route are calculated based on a single traversal, then multiplied by the annual number of trips. If the time to traverse a route increases, this increases driver costs and the share of annualized vehicle cost that is allocated to the route. If the cargo capacity per trip changes, the annual cost of operating the route also changes.

Response regarding axle configurations: If we understand the reviewer correctly, they assume that all electric trucks are 4x2s (note to the editor: this means a truck with four wheels, with propulsion on two) and that it is therefore not enough to only consider changes in gross train weight, but we also need to consider how a potential change in wheel configuration would affect the weight limits. As this was a bit beyond our expertise, we reached out to contacts from Scania and Volvo for a better understanding. They stated that most tractor-trailer combinations in Europe use 4x2 configurations, to minimize the turning radius, while 6x4 is a more common configuration in North America. The initial BET offers in the European markets are therefore 4x2, but neither manufacturer expects any future difference in product offerings vs diesel. There should be no need to switch to a different wheel configuration when switching to electric, which we hope will settle the reviewer’s concern. Manufacturers including Scania, Volvo and Tesla have electric 6x4 trucks in operation today, but we have not been able to confirm if these configurations have yet reached series production.

Comment 3: *“Papers such as <https://doi.org/10.1109/ITSC57777.2023.10422258> suggest that it may be beneficial to have split rest stops compared to one continuous rest stop as is also permitted in EU regulations. Changes in rest stop regulations will affect battery sizes and charger usage and should be addressed by the model.”*

Response regarding stop frequency: We had not previously come across the paper mentioned by the reviewer, which was published online until our paper was initially submitted for review. Since it’s an interesting finding, we verified the robustness of our own findings by replicating the recommended study in MOSTACHI. Additional simulations were run using the “Neutral” condition in the paper with 0, 2000 and 6000 km of policy-supported ERS, with and without shortened stops. This is described in the manuscript at the end of “Demand for Batteries and Combustion Engine Fuels” and in Supplementary figure 5.

Our simulations confirm the findings from the prior research and reveal that the effect stacks on top of the ERS-induced effects that we describe in the rest of the paper. We have not been able to identify any case where frequent stopping changes the qualitative effects we have identified. We therefore argue the prior work does not present an obstacle to publishing this research.

Comment 4: *“Along with improvements in battery technology, improvements in road and tyre technology may result in higher axle load limit allowances, which would mean smaller payload losses and larger batteries possible in trucks.”*

Response regarding increased weight limits for electric trucks: We understand this comment as referring to the proposal to raise the maximum gross train weight of electric trucks to be more than

that of (the same class of) diesel trucks. This would allow electric trucks with high-capacity battery packs to carry the same amount of cargo as diesel trucks that have a lower empty weight. Road authorities have so far been resistant to permit this, as they need to guarantee the durability of the roads.

What we observe in our experiments is that there is a preference for electric trucks to charge dynamically via electric road systems, as this enables the electric trucks to conduct today's operations also with small battery packs. An electric truck with smaller battery pack will always cost less to purchase and weight less (thus have higher earning potential and/or lower fuel consumption) than an electric truck with larger battery pack. We therefore cannot see how the proposed change would strengthen the competitiveness of static charging versus dynamic charging.

In the absence of ERS, the proposed change could matter. Greater permitted carrying capacity for BETs would strengthen their competitiveness versus ICETs. Batteries are too expensive for this to be an attractive solution in early model years, but such a policy change could have impact beyond 2040. There are also many other policy implementations that could give BETs competitive advantages over ICETs. We think another study would be required to analyze such effects.

Comment 5.1: *“The energy consumption values (kWh/km) for all categories of trucks do not seem to be dependent on the payload as seen in the supplementary information table. These values would be different for an empty and loaded truck (in the same category) and would be dependent on the vehicle’s mass.”*

Response regarding weight-dependent energy consumption: We made energy consumption dependent on changes in empty vehicle weight in Revision 1. This change in the manuscript was only highlighted in an MS Word comment, which we believe was stripped out in the submission process and thus very difficult to notice. The model implementation is now that a ten percent change in gross train weight leads to a five percent change in energy consumption, based on cited measurements. This is explained in *Method – Model Calculations – Weight adjustments*. In this revision, mentions of the energy consumption adjustments have also been added to the flowchart in Figure 8 and in Supplementary table 7.

The reviewer is correct in pointing out that we do not vary energy consumption based on per-route cargo fill rates. The route data we had access to did not make it straightforward to do this and it is left (and proposed) as future work. We cannot see why this simplification would invalidate any of our findings, but if MOSTACHI is used in the future to plan precise placement of charging stations, such an extension would be recommended.

Comment 5.2: *“Furthermore, it is assumed that they reduce over time (is it by 1% every year)? This assumption might make sense for ICEVs as powertrain efficiency keeps increasing (as they are not very efficient currently ~40%), but BEV powertrains are already around 90% efficient... hence, similar efficiency improvements may not be expected. What may improve is the energy density of batteries, resulting in smaller battery sizes and reduced weights and theoretically reduced energy consumption, but it wouldn't in reality as operators would then carry more payload up to the weight limit to reduce logistics cost.”*

Response regarding powertrain efficiency improvements: The reviewer makes a good point, and we have updated the assumptions so that fuel consumption for ICETs decreases by 0.5% per year and energy consumption for BETs decreases by 0.6% per year. General improvements to rolling resistance and aerodynamics (0.5%/year) are assumed to be shared for all powertrains. ICEs are not assumed to get more efficient due to lack of R&D investments. A modest additional 0.1%/year improvement is assumed for trucks with battery electric powertrains, estimated based on that (the

already very low) heat losses decrease at the same relative rate as they have done historically for ICEs.

Comment 5.3: *“Additionally, the energy consumption depends highly on the route taken because of factors like road gradients and traffic. All these assumptions should be reconsidered as the study is very sensitive to energy consumption.”*

Response: Both road gradients and traffic are now explicitly discussed in *Method – Model Capabilities and Limitations*. We would like to reiterate that the scientific contribution of the paper is to understand the marginal effects of introducing ERS in parallel with static charging, not to provide perfect estimates of the energy consumption of a population of electric trucks or to perfectly optimize placement of charging infrastructure. As the reviewer points out, factors that affect energy consumption can indeed affect the results, but we argue that the primary focus needs to be on modelling of factors that affect electric vehicles *differently* depending on how they are charged. We believe we have captured the main factors that affect this *difference*.

- 1) **Road gradients:** We are in contact with a research group in Canada which is currently working on adding support for road gradient-dependent energy consumption in MOSTACHI. This group has prior experience studying the impact of road gradient on truck energy consumption. After discussing the topic with Ken Darcovich, a member of the Canadian group, he confirmed that energy consumption does increase in hilly terrain, with greater losses for diesel vehicles than for electric vehicles, as electric vehicles can regenerate potential energy when going downhill. This means that road gradient would need to be considered for accurate estimates of overall electrification rates if MOSTACHI was to be applied to a mountainous geography. Sweden, however, is generally too flat for this to matter. Ken estimated that we would not see more than 5% difference in total fuel consumption along the main national transport routes and differences between powertrains below that. Differences between electric trucks using different modes of charging would be significantly smaller.
- 2) **Traffic:** The main impact traffic has on the difference in utility provided by static and dynamic charging is that denser traffic favors charging solutions with greater economies of scale (ERS), and conversely, that ERS requires a critical mass of users to become an economically viable alternative. MOSTACHI's handling of this effect is one of its main advantages over previous research methods – charging behaviors are optimized along individual routes and summed across routes, accounting levelized utilization costs for charging infrastructure, which are fed back iteratively to influence charging patterns, until the system converges.

MOSTACHI is however not a traffic simulation tool (such as MATSim), capable of understanding for instance when and where traffic jams are likely to occur, how traffic congestion may affect or be affected by charging infrastructure utilization, or how congestion at chargers may impact the efficiency of road freight. We argue that it is unrealistic to expect this level of granularity from a simulation at national scale over decades and that such questions are better answered using other tools and experimental designs. However, we do agree that such complementary research is important, in particular when planning networks of public static fast charging for commercial vehicles, as the physical distance between mandated driver rest stops decreases in the presence of traffic jams. From this perspective, planning of ERS infrastructure is less complex.

Comment 6: *“Rather than directly having the Results section after the Introduction, it is suggested to split it as Introduction, Research Questions, Methodology (not the full MOSTACHI explanation, but just an explanation of how MOSTACHI was used or set up – perhaps the Experiment Design bit) and then the Results/Discussion, followed finally by the current ‘Methods’ section which could also be renamed to ‘Details of the Simulation Tool (MOSTACHI)’ or something similar. This would*

probably make it easier to go through as seeing the Results after the Introduction and Methods at the end is undeniably hard to figure out.”

Response regarding document structure: Our understanding of the Nature Communication guidelines is that the current ordering of the sections is required by the journal, while the suggested changes would not comply with the guidelines. We have therefore not made the proposed changes.

RESPONSE LETTER

REVISION 3

All changes have been tracked in the manuscript.

CHANGES IN REVISION 3 TO COMPLY WITH THE JOURNAL'S GUIDELINES

1. The title and author list were moved to a separate page. Contact information was added.
2. The title of the manuscript was changed to "*Correcting market failure for no-regret electric road investments under uncertainty*". The previous title contained punctuation (:) in place of an active verb.
3. Line spacing was changed to double.
4. Page numbering was moved to bottom-center.
5. All bullet and numbered lists in the manuscript were converted to body text (with no changes in meaning).
6. All figures and tables, with captions, were moved to separate sections at the end of the manuscript. The caption for table 1 was moved before the table. Figure references were changed from "Figure N" to "Fig. N".
7. The section "Code and Data Availability" was split into two sections.
8. Algorithm 1 was moved to the supplementary material, now Supplementary Method 1.
9. Line breaks were removed in the section Author Contributions.
10. A cover page and Supplementary Items List was added to the supplementary information.
11. An additional data file in Excel format was created with data structured to more easily recreate all data figures. Recreation of the figures is likely not possible in Excel.

REVIEWER 1

No requests for changes. We thank the reviewer for their constructive feedback that has helped us improve the quality of the manuscript.

REVIEWER 3

Comment 1: "*Regarding range-extended vehicles: It would be good to be able to simulate those journeys away from charging infrastructure, particularly in countries with such journeys such as Australia. This could be added in a future update to the software so that it is capable of simulating more countries.*"

Response: The following clarification was added to the Model Limitations section: "*The omission of hybrids includes BETs with temporarily installed range extending auxiliary power units.*"

Comment 2: "*Regarding axle configuration: Currently the highest weight category (44T) vehicles are 6x2 vehicles. In BEVs, manufacturers place batteries between the front and rear axles of the tractor unit. This takes up some space in the wheelbase, which means that the length of a 4x2 BEV would be the same as a 6x2 ICE truck (16.5 metres). There is a maximum limit on the combination length (tractor+trailer), which is 16.5 metres. A 6x2 (or 6x4) BEV exceeds this limit due to the increase in the wheelbase of the tractor caused by the battery packs. Therefore, under current regulations, it is not possible to use a 6x2 or 6x4 articulated BEV on the road with a standard trailer even if manufacturers may have made such trucks. This is important to account for as it would need a major change in regulations to allow this to happen.*"

Response: We have added the following text to the Model Limitations section: "*the model does not incorporate current legal constraints on axle configuration or vehicle length. Battery packs above roughly 600 kWh usable (or more in the future, with improved energy density) increase axle load,*

particularly in 4x2 configurations. To stay within axle limits, a shift to 6x2 or 6x4 layouts may be required, which reduces available space for batteries unless vehicle length increases. Simulated trucks with very high battery capacity may therefore exceed length limits for articulated tractor-trailer combinations unless paired with shorter trailers or granted legal exceptions.”

A brief caveat was included in the discussion of figure 3.

We have also added the following text to the Discussion section: “Our simulation model does not constrain vehicle configurations by axle layout or legal length. In practice, cost-minimizing battery capacities identified in scenarios without ERS – typically 700 kWh and above – may not be physically compatible with axle load limits of current roads and with the EU’s legal length limits for tractor-trailer combinations. Raising length or axle limits is possible and could ease electrification while negatively affecting turning radiuses, safety, and infrastructure compatibility. Capping battery pack capacity around 700 kWh would significantly reduce simulated BET uptake in scenarios without ERS and would make ERS investments less risky in scenarios without supportive policies. By enabling smaller battery packs, ERS-equipped trucks would be more likely to stay within historical limits, avoiding such trade-offs. Even if longer and heavier vehicles are permitted, smaller battery packs leave more capacity for cargo.

CHANGES IN REVISION 4 OF NCOMMS-23-46159E

1. An additional supplementary figure (Sup. Fig. 4) was added. This figure helped me improve my own understanding of some of the marginal effects of increasing charging infrastructure availability in the simulation and I thought it could be useful also for others who try to assess the correctness and implications of the work. The new figure uses the same data as in supplementary figure 3, but the plot shows differences on the Y axis rather than absolute values.